# Data-dependent Sample Complexity of Deep Neural Networks via Lipschitz Augmentation

**Colin Wei**
Computer Science Department
Stanford University
colinwei@stanford.edu

**Tengyu Ma**
Computer Science Department
Stanford University
tengyuma@stanford.edu

## Abstract

Existing Rademacher complexity bounds for neural networks rely only on norm control of the weight matrices and depend exponentially on depth via a product of the matrix norms. Lower bounds show that this exponential dependence on depth is unavoidable when no additional properties of the training data are considered. We suspect that this conundrum comes from the fact that these bounds depend on the training data only through the margin. In practice, many *data-dependent* techniques such as Batchnorm improve the generalization performance. For feedforward neural nets as well as RNNs, we obtain tighter Rademacher complexity bounds by considering additional data-dependent properties of the network: the norms of the hidden layers of the network, and the norms of the Jacobians of each layer with respect to all previous layers. Our bounds scale polynomially in depth when these empirical quantities are small, as is usually the case in practice. To obtain these bounds, we develop general tools for augmenting a sequence of functions to make their composition Lipschitz and then covering the augmented functions. Inspired by our theory, we directly regularize the network's Jacobians during training and empirically demonstrate that this improves test performance.

## 1 Introduction

Deep networks trained in practice typically use many more parameters than training examples, and therefore have the capacity to overfit to the training set [Zhang et al., 2016]. Fortunately, there are also many known (and unknown) sources of regularization during training: model capacity regularization such as simple weight decay, implicit or algorithmic regularization [Gunasekar et al., 2017, 2018b, Soudry et al., 2018, Li et al., 2018], and finally regularization that depends on the training data such as Batchnorm [Ioffe and Szegedy, 2015], layer normalization [Ba et al., 2016], group normalization [Wu and He, 2018], path normalization [Neyshabur et al., 2015a], dropout [Srivastava et al., 2014, Wager et al., 2013], and regularizing the variance of activations [Littwin and Wolf, 2018].

In many cases, it remains unclear why data-dependent regularization can improve the final test error — for example, why Batchnorm empirically improves the generalization performance in practice [Ioffe and Szegedy, 2015, Zhang et al., 2019]. We do not have many tools for analyzing data-dependent regularization in the literature; with the exception of Dziugaite and Roy [2018], [Arora et al., 2018] and [Nagarajan and Kolter, 2019] (with which we compare later in more detail), existing bounds typically consider properties of the weights of the learned model but little about their interactions with the training set. Formally, define a data-dependent property as any function of the learned model and the training data. In this work, we prove tighter generalization bounds by considering additional data-dependent properties of the network. Optimizing these bounds leads to data-dependent regularization techniques that empirically improve performance.

One well-understood and important data-dependent property is the training margin: Bartlett et al. [2017] show that networks with larger normalized margins have better generalization guarantees. However, neural nets are complex, so there remain many other data-dependent properties which could potentially lead to better generalization. We extend the bounds and techniques of Bartlett et al. [2017] by considering additional properties: the hidden layer norms and interlayer Jacobian norms. Our final generalization bound (Theorem 5.1) is a polynomial in the hidden layer norms and Lipschitz constants on the training data. We give a simplified version below for expositional purposes. Let $F$ denote a neural network with *smooth* activation $\phi$ parameterized by weight matrices $\{W^{(i)}\}_{i=1}^r$ that perfectly classifies the training data with margin $\gamma > 0$. Let $t$ denote the maximum $\ell_2$ norm of any hidden layer or training datapoint, and $\sigma$ the maximum operator norm of any interlayer Jacobian, where both quantities are evaluated *only on the training data*.

**Theorem 1.1** (Simplified version of Theorem 5.1). *Suppose $\sigma, t \geq 1$. With probability $1 - \delta$ over the training data, we can bound the test error of $F$ by*

$$L_{0\text{-}1}(F) \leq \widetilde{O}\left( \frac{(\frac{\sigma}{\gamma} + r^3\sigma^2)t\left(1 + \sum_i \|W^{(i)\top}\|_{2,1}^{2/3}\right)^{3/2} + r^2\sigma\left(1 + \sum_i \|W^{(i)}\|_{1,1}^{2/3}\right)^{3/2}}{\sqrt{n}} + r\sqrt{\frac{\log(\frac{1}{\delta})}{n}} \right)$$

*The notation $\widetilde{O}$ hides logarithmic factors in $d, r, \sigma, t$ and the matrix norms. The $\|\cdot\|_{2,1}$ norm is formally defined in Section 3.*

The degree of the dependencies on $\sigma$ may look unconventional — this is mostly due to the dramatic simplification from our full Theorem 5.1, which obtains a more natural bound that considers all interlayer Jacobian norms instead of only the maximum. Our bound is polynomial in $t, \sigma$, and network depth, but independent of width. In practice, $t$ and $\sigma$ have been observed to be much smaller than the product of matrix norms [Arora et al., 2018, Nagarajan and Kolter, 2019]. We remark that our bound is not homogeneous because the smooth activations are not homogeneous and can cause a second order effect on the network outputs.

In contrast, the bounds of Neyshabur et al. [2015b], Bartlett et al. [2017], Neyshabur et al. [2017a], Golowich et al. [2017] all depend on a product of norms of weight matrices which scales exponentially in the network depth, and which can be thought of as a worst case Lipschitz constant of the network. In fact, lower bounds show that with only norm-based constraints on the hypothesis class, this product of norms is unavoidable for Rademacher complexity-based approaches (see for example Theorem 3.4 of [Bartlett et al., 2017] and Theorem 7 of [Golowich et al., 2017]). We circumvent these lower bounds by additionally considering the model's Jacobian norms – empirical Lipschitz constants which are much smaller than the product of norms because they are only computed on the training data.

The bound of Arora et al. [2018] depends on similar quantities related to noise stability but only holds for a compressed network and not the original. The bound of Nagarajan and Kolter [2019] also depends polynomially on the Jacobian norms rather than exponentially in depth; however these bounds also require that the inputs to the activation layers are bounded away from 0, an assumption that does not hold in practice [Nagarajan and Kolter, 2019]. We do not require this assumption because we consider networks with smooth activations, whereas the bound of Nagarajan and Kolter [2019] applies to relu nets.

In Section G, we additionally present a generalization bound for recurrent neural nets that scales polynomially in the same quantities as our bound for standard neural nets. Prior generalization bounds for RNNs either require parameter counting [Koiran and Sontag, 1997] or depend exponentially on depth [Zhang et al., 2018, Chen et al., 2019].

In Figure 1, we plot the distribution over the sum of products of Jacobian and hidden layer norms (which is the leading term of the bound in our full Theorem 5.1) for a WideResNet [Zagoruyko and Komodakis, 2016] trained with and without Batchnorm. Figure 1 shows that this sum blows up for networks trained without Batchnorm, indicating that the terms in our bound are empirically relevant for explaining data-dependent regularization.

An immediate bottleneck in proving Theorem 1.1 is that standard tools require fixing the hypothesis class before looking at training data, whereas conditioning on data-dependent properties makes the hypothesis class a random object depending on the data. A natural attempt is to augment the loss

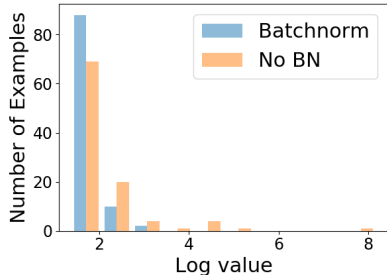

Figure 1: Let $h_1, h_2, h_3$ denote the 1st, 2nd, and 3rd blocks of a 16-layer WideResNet and $J_i$ the Jacobian of the output w.r.t layer $i$. In log-scale we plot a histogram of the 100 largest values on the training set of $\sum_{i=1}^{3} \|h_i\|\|J_i\|/\gamma$ for a WideResNet trained with and without Batchnorm on CIFAR10, where $\gamma$ is the example's margin.

with indicators on the intended data-dependent quantities $\{\gamma_i\}$, with desired bounds $\{\kappa_i\}$ as follows:

$$l_{\text{aug}} = (l_{\text{old}} - 1) \prod_{\text{properties } \gamma_i} \mathbb{1}(\gamma_i \leq \kappa_i) + 1$$

This augmented loss upper bounds the original loss $l_{\text{old}} \in [0, 1]$, with equality when all properties hold for the training data. The augmentation lets us reason about a hypothesis class that is independent of the data by directly conditioning on data-dependent properties in the loss. The main challenges with this approach are twofold: 1) designing the correct set of properties and 2) proving generalization of the final loss $l_{\text{aug}}$, a complicated function of the network.

Our main tool is covering numbers: Lemma 4.1 shows that a composition of functions (i.e, a neural network) has low covering number if the output is worst-case Lipschitz at each level of the composition and internal layers are bounded in norm. Unfortunately, the standard neural net loss satisfies neither of these properties (without exponential dependencies on depth). However, by augmenting with properties $\gamma$, we can guarantee they hold. One technical challenge is that augmenting the loss makes it harder to reason about covering, as the indicators can introduce complicated dependencies between layers.

Our main technical contributions are: 1) We demonstrate how to augment a composition of functions to make it Lipschitz at all layers, and thus easy to cover. Before this augmentation, the Lipschitz constant could scale exponentially in depth (Theorem 4.4). 2) We reduce covering a complicated sequence of operations to covering the individual operations (Theorem 4.3). 3) By combining 1 and 2, it follows cleanly that our augmented loss on neural networks has low covering number and therefore has good generalization. Our bound scales polynomially, not exponentially, in the depth of the network when the network has good Lipschitz constants on the training data (Theorem 5.1).

As a complement to the main theoretical results in this paper, we show empirically in Section 6 that directly regularizing our complexity measure can result in improved test performance.

## 2 Related Work

Zhang et al. [2016] and Neyshabur et al. [2017b] show that generalizaton in deep learning often disobeys conventional statistical wisdom. One of the approaches adopted torwards explaining generalization is implicit regularization; numerous recent works have shown that the training method prefers minimum norm or maximum margin solutions [Soudry et al., 2018, Li et al., 2018, Ji and Telgarsky, 2018, Gunasekar et al., 2017, 2018a,b, Wei et al., 2018]. With the exception of [Wei et al., 2018], these papers analyze simplified settings and do not apply to larger neural networks.

This paper more closely follows a line of work related to Rademacher complexity bounds for neural networks [Neyshabur et al., 2015b, 2018, Bartlett et al., 2017, Golowich et al., 2017]. For a comparison, see the introduction. There has also been work on deriving PAC-Bayesian bounds for generalization [Neyshabur et al., 2017b,a, Nagarajan and Kolter, 2019]. Dziugaite and Roy [2017a] optimize a bound to compute non-vacuous bounds for generalization error. Another line of work analyzes neural nets via their behavior on noisy inputs. Neyshabur et al. [2017b] prove PAC-Bayesian generalization bounds for random networks under assumptions on the network's empirical noise stability. Arora et al. [2018] develop a notion of noise stability that allows for compression of a network under an appropriate noise distribution. They additionally prove that the compressed network generalizes well. In comparison, our Lipschitzness construction also relates to noise stability, but our bounds hold for the original network and do not rely on the particular noise distribution.

Nagarajan and Kolter [2019] use PAC-Bayes bounds to prove a similar result as ours for generalization of a network with bounded hidden layer and Jacobian norms. The main difference is that their bounds depend on the inverse relu preactivations, which are found to be large in practice [Nagarajan and Kolter, 2019]; our bounds apply to smooth activations and avoid this dependence at the cost of an additional factor in the Jacobian norm (shown to be empirically small). We note that the choice of smooth activations is empirically justified [Clevert et al., 2015, Klambauer et al., 2017]. We also work with Rademacher complexity and covering numbers instead of the PAC-Bayes framework. It is relatively simple to adapt our techniques to relu networks to produce a similar result to that of Nagarajan and Kolter [2019], by conditioning on large pre-activation values in our Lipschitz augmentation step (see Section 4.2). In Section H, we provide a sketch of this argument and obtain a bound for relu networks that is polynomial in hidden layer and Jacobian norms and inverse preactivations. However, it is not obvious how to adapt the argument of Nagarajan and Kolter [2019] to activation functions whose derivatives are not piecewise-constant.

Dziugaite and Roy [2018, 2017b] develop PAC-Bayes bounds for data-dependent priors obtained via some differentially private mechanism. Their bounds are for a randomized classifier sampled from the prior, whereas we analyze a deterministic, fixed model.

Novak et al. [2018] empirically demonstrate that the sensitivity of a neural net to input noise correlates with generalization. Sokolić et al. [2017], Krueger and Memisevic [2015] propose stability-based regularizers for neural nets. Hardt et al. [2015] show that models which train faster tend to generalize better. Keskar et al. [2016], Hoffer et al. [2017] study the effect of batch size on generalization. Brutzkus et al. [2017] analyze a neural network trained on hinge loss and linearly separable data and show that gradient descent recovers the exact separating hyperplane.

## 3 Notation

Let $\mathbb{1}(E)$ be the indicator function of event $E$. Let $l_{0\text{-}1}$ denote the standard 0-1 loss. For $\kappa \geq 0$, Let $\mathbb{1}_{\leq \kappa}(\cdot)$ be the softened indicator function defined as

$$
\mathbb{1}_{\leq \kappa}(t) = \begin{cases} 1 & \text{if } t \leq \kappa \\ 2 - t/\kappa & \text{if } \kappa \leq t \leq 2\kappa \\ 0 & \text{if } 2\kappa \leq t \end{cases}
$$

Note that $\mathbb{1}_{\leq \kappa}$ is $\kappa^{-1}$-Lipschitz. Define the norm $\| \cdot \|_{p,q}$ by $\|A\|_{p,q} \triangleq \left( \sum_j \left( \sum_i A_{i,j}^p \right)^{q/p} \right)^{1/q}$. Let $P_n$ be a uniform distribution over $n$ points $\{x_1, \ldots, x_n\} \subset \mathcal{D}_x$. Let $f$ be a function that maps $\mathcal{D}_x$ to some output space $\mathcal{D}_f$, and assume both spaces are equipped with some norms $\|\!|\cdot|\!\|$ (these norms can be different but we use the same notations for them). Then the $L_2(P_n, \|\!|\cdot|\!\|)$ norm of the function $f$ is defined as $\|f\|_{L_2(P_n,\|\!|\cdot|\!\|)} \triangleq \left( \frac{1}{n} \sum_i \|\!|f(x_i)|\!\|^2 \right)^{1/2}$. We use $D$ to denote total derivative operator, and thus $Df(x)$ represents the Jacobian of $f$ at $x$. Suppose $\mathcal{F}$ is a family of functions from $\mathcal{D}_x$ to $\mathcal{D}_f$. Let $\mathcal{C}(\epsilon, \mathcal{F}, \rho)$ be the covering number of the function class $\mathcal{F}$ w.r.t. metric $\rho$ with cover size $\epsilon$. In many cases, the covering number depends on the examples through the norms of the examples, and in this paper we only work with these cases. Thus, we let $\mathcal{N}(\epsilon, \mathcal{F}, s)$ be the maximum covering number for any possible $n$ data points with norm not larger than $s$. Precisely, if we define $\mathcal{P}_{n,s}$ to be the set of all possible uniform distributions supported on $n$ data points with norms not larger than $s$, then $\mathcal{N}(\epsilon, \mathcal{F}, s) \triangleq \sup_{P_n \in \mathcal{P}_{n,s}} \mathcal{C}(\epsilon, \mathcal{F}, L_2(P_n, \|\!|\cdot|\!\|))$. Suppose $\mathcal{F}$ contains functions with $m$ inputs that map from a tensor product $m$ Euclidean space to Euclidean space, then we define $\mathcal{N}(\epsilon, \mathcal{F}, (s_1, \ldots, s_m)) \triangleq \sup_{\substack{P:\forall (x_1,\ldots,x_m) \in \text{supp}(P) \\ \|x_i\| \leq s_i}} \mathcal{C}(\epsilon, \mathcal{F}, L_2(P))$.

## 4 Overview of Main Results and Proof Techniques

In this section, we give a general overview of the main technical results and outline how to prove them with minimal notation. We will point to later sections where many statements are formalized.

To simplify the core mathematical reasoning, we abstract feed-forward neural networks (including residual networks) as compositions of operations. Let $\mathcal{F}_1, \ldots, \mathcal{F}_k$ be a sequence of families of functions (corresponding to families of single layer neural nets in the deep learning setting) and $\ell$ be

a Lipschitz loss function taking values in $[0, 1]$. We study the compositions of $\ell$ and functions in $\mathcal{F}_i$'s:

$$\mathcal{L} \triangleq \ell \circ \mathcal{F}_k \circ \mathcal{F}_{k-1} \cdots \circ \mathcal{F}_1 = \{\ell \circ f_k \circ f_{k-1} \circ \cdots \circ f_1 : \forall i, f_i \in \mathcal{F}_i\} \tag{1}$$

Textbook results [Bartlett and Mendelson, 2002] bound the generalization error by the Rademacher complexity (formally defined in Section C) of the family of losses $\mathcal{L}$, which in turn is bounded by the covering number of $\mathcal{L}$ through Dudley's entropy integral theorem [Dudley, 1967]. Modulo minor nuances, the key remaining question is to give a tight covering number bound for the family $\mathcal{L}$ for every target cover size $\epsilon$ in a certain range (often, considering $\epsilon \in [1/n^{O(1)}, 1]$ suffices).

As alluded to in the introduction, generalization error bounds obtained through this machinery only depend on the (training) data through the margin in the loss function, and our aim is to utilize more data-dependent properties. Towards understanding which data-dependent properties are useful to regularize, it is helpful to revisit the data-independent covering technique of [Bartlett et al., 2017], the skeleton of which is summarized below.

Recall that $\mathcal{N}(\epsilon, \mathcal{F}, s)$ denotes the covering number for arbitrary $n$ data points with norm less than $s$. The following lemma says that if the intermediate variable (or the hidden layer) $f_i \circ \cdots \circ f_1(x)$ is bounded, and the composition of the rest of the functions $l \circ f_k \circ \cdots \circ f_{i+1}(x)$ is Lipschitz, then small covering number of local functions imply small covering number for the composition of functions.

**Lemma 4.1.** *[abstraction of techniques in [Bartlett et al., 2017]] In the context above, assume:*

1. *for any $x \in supp(P_n)$, $\|\|f_i \circ \cdots \circ f_1(x)\|\| \le s_i$.*

2. *$\ell \circ f_k \circ \cdots \circ f_{i+1}$ is $\kappa_i$-Lipschitz for all $i$.*

*Then, we have the following covering number bound for $\mathcal{L}$ (for any choice of $\epsilon_1, \ldots, \epsilon_k > 0$):*
$\log \mathcal{N}(\sum_{i=1}^{k} \kappa_i \epsilon_i, \mathcal{L}, s_0) \le \sum_{i=1}^{k} \log \mathcal{N}(\epsilon_i, \mathcal{F}_i, s_{i-1})$.

The lemma says that the log covering number and the cover size scale linearly if the Lipschitzness parameters and norms remain constant. However, these two quantities, in the worst case, can easily scale exponentially in the number of layers, and they are the main sources of the dependency of product of spectral/Frobenius norms of layers in [Golowich et al., 2017, Bartlett et al., 2017, Neyshabur et al., 2017a, 2015b] More precisely, the worst-case Lipschitzness over all possible data points can be exponentially bigger than the average/typical Lipschitzness for examples randomly drawn from the training or test distribution. We aim to bridge this gap by deriving a generalization error bound that only depends on the Lipschitzness and boundedness on the training examples.

Our general approach, partially inspired by margin theory, is to augment the loss function by soft indicators of Lipschitzness and boundedness. Let $h_i$ be shorthand notation for $f_i \circ \cdots \circ f_1$, the $i$-th intermediate value, and let $z(x) \triangleq \ell(h_k(x))$ be the original loss. Our first attempt considered:

$$\tilde{z}'(x) \triangleq 1 + (z(x) - 1) \cdot \prod_{i=1}^{k} \mathbb{1}_{\le s_i}(\|h_i(x)\|) \cdot \prod_{i=1}^{k} \mathbb{1}_{\le \kappa_i}(\|\partial z / \partial h_i\|_{\text{op}}) \tag{2}$$

Since $z$ takes values in $[0, 1]$, the augmented loss $\tilde{z}'$ is an upper bound on the original loss $z$ with equality when all the indicators are satisfied with value 1. The hope was that the indicators would flatten those regions where $h_i$ is not bounded and where $z$ is not Lipschitz in $h_i$. However, there are two immediate issues. First, the soft indicators functions are themselves functions of $h_i$. It's unclear whether the augmented function can be Lipschitz with a small constant w.r.t $h_i$, and thus we cannot apply Lemma 4.1.[1] Second, the augmented loss function becomes complicated and doesn't fall into the sequential computation form of Lemma 4.1, and therefore even if Lipschitzness is not an issue, we need new covering techniques beyond Lemma 4.1.

We address the first issue by *recursively* augmenting the loss function by multiplying more soft indicators that bound the Jacobian of the current function. The final loss $\tilde{z}$ reads:[2]

$$\tilde{z}(x) \triangleq 1 + (z(x) - 1) \cdot \prod_{i=1}^{k} \mathbb{1}_{\le s_i}(\|h_i(x)\|) \cdot \prod_{1 \le i \le j \le k} \mathbb{1}_{\le \kappa_{j \leftarrow i}}(\|Df_j \circ \cdots \circ f_i[h_{i-1}]\|_{\text{op}}) \tag{3}$$

where $\kappa_{j \leftarrow i}$'s are user-defined parameters. For our application to neural nets, we instantiate $s_i$ as the maximum norm of layer $i$ and $\kappa_{j \leftarrow i}$ as the maximum norm of the Jacobian between layer $j$ and $i$ across the training dataset. A polynomial in $\kappa, s$ can be shown to bound the worst-case Lipschitzness of the function w.r.t. the intermediate variables in the formula above.[3] By our choice of $\kappa, s$, a) the training loss is unaffected by the augmentation and b) the worst-case Lipschitzness of the loss is controlled by a polynomial of the Lipschitzness on the training examples. We provide an informal overview of our augmentation procedure in Section 4.2 and formally state definitions and guarantees in Section B. The downside of the Lipschitz augmentation is that it further complicates the loss function. Towards covering the loss function (assuming Lipschitz properties) efficiently, we extend Lemma 4.1, which works for sequential compositions of functions, to general families of formulas, or computational graphs. We informally overview this extension in Section 4.1 using a minimal set of notations, and in Section A, we give a formal presentation of these results.

Combining the Lipschitz augmentation and graphs covering results, we obtain a covering number bound of augmented loss. The theorem below is formally stated in Theorem B.3 of Section B.

**Theorem 4.2.** *Let $\tilde{\mathcal{L}}$ be the family of augmented losses defined in (3). For cover resolutions $\epsilon_i$ and values $\tilde{\kappa}_i$ that are polynomial in the parameters $s_i, \kappa_{j \leftarrow i}$, we obtain the following covering number bound for $\tilde{\mathcal{L}}$:*

$$\log \mathcal{N}(\sum_i \epsilon_i \tilde{\kappa}_i, \tilde{\mathcal{L}}, s_0) \leq \sum_i \log \mathcal{N}(\epsilon_i, \mathcal{F}_i, s_{i-1}) + \sum_i \log \mathcal{N}(\epsilon_i, D\mathcal{F}_i, s_{i-1})$$

*where $D\mathcal{F}_i$ denotes the function class obtained from applying the total derivative operator to all functions in $\mathcal{F}_i$.*

Now, following the standard technique of bounding Rademacher complexity via covering numbers, we can obtain generalization error bounds for augmented loss. For the demonstration of our technique, suppose that the following simplification holds: $\log \mathcal{N}(\epsilon_i, D\mathcal{F}_i, s_{i-1}) = \log \mathcal{N}(\epsilon_i, \mathcal{F}_i, s_{i-1}) = s_{i-1}^2/\epsilon_i^2$. Then after minimizing the covering number bound in $\epsilon_i$ via standard techniques, we obtain the below generalization error bound on the original loss for parameters $\tilde{\kappa}_i$ alluded to in Theorem 4.2 and formally defined in Theorem B.2. When the training examples satisfy the augmented indicators, $\mathbb{E}_{\text{train}}[\tilde{z}] = \mathbb{E}_{\text{train}}[z]$, and because $\tilde{z}$ bounds $z$ from above, we have

$$\mathbb{E}_{\text{test}}[z] - \mathbb{E}_{\text{train}}[z] \leq \mathbb{E}_{\text{test}}[\tilde{z}] - \mathbb{E}_{\text{train}}[\tilde{z}] \leq \tilde{O}\left( \frac{\left(\sum_i \tilde{\kappa}_i^{2/3} s_{i-1}^{2/3}\right)^{3/2}}{\sqrt{n}} + \sqrt{\frac{\log(1/\delta)}{n}} \right) \tag{4}$$

## 4.1 Overview of Computational Graph Covering

To obtain the augmented $\tilde{z}$ defined in (3), we needed to condition on data-dependent properties which introduced dependencies between the various layers. Because of this, Lemma 4.1 is no longer sufficient to cover $\tilde{z}$. In this section, we informally overview how to extend Lemma 4.1 to cover more general functions via the notion of computational graphs. *For space constraints, this section is a dramatically abbreviated and informal version of Section A.*

A computational graph $G(\mathcal{V}, \mathcal{E}, \{R_V\})$ is an acyclic directed graph with three components: the set of nodes $\mathcal{V}$ corresponds to variables, the set of edges $\mathcal{E}$ describes dependencies between these variables, and $\{R_V\}$ contains a list of composition rules indexed by the variables $V$'s, representing the process of computing $V$ from its direct predecessors. For simplicity, we assume the graph contains a unique sink, denoted by $O_G$, and we call it the "output node". We also overload the notation $O_G$ to denote the function that the computational graph $G$ finally computes. Let $\mathcal{I}_G = \{I_1, \dots, I_p\}$ be the subset of nodes with no predecessors, which we call the "input nodes" of the graph.

The notion of a family of computational graphs generalizes the sequential family of function compositions in (1). Let $\mathcal{G} = \{G(\mathcal{V}, \mathcal{E}, \{R_V\})\}$ be a family of computational graphs with shared nodes, edges, output node, and input nodes (denoted by $\mathcal{I}$). Let $\mathfrak{R}_V$ be the collection of all possible composition rules used for node $V$ by the graphs in the family $\mathcal{G}$. This family $\mathcal{G}$ defines a set of functions $O_{\mathcal{G}} \triangleq \{O_G : G \in \mathcal{G}\}$.

The theorem below extends Lemma 4.1. In the computational graph interpretation, Lemma 4.1 applies to a sequential family of computational graphs with $k$ internal nodes $V_1, \ldots, V_k$, where each $V_i$ computes the function $f_i$, and the output computes the composition $O_G = \ell \circ f_k \cdots \circ f_1 = z$. However, the augmented loss $\tilde{z}$ no longer has this sequential structure, requiring the below theorem for covering generic families of computational graphs. We show that covering a general family of computational graphs can be reduced to covering all the local composition rules.

**Theorem 4.3** (Informal and weaker version of Theorem A.3). *Suppose that there is an ordering $(V_1, \ldots, V_m)$ of the nodes, so that after cutting out nodes $V_1, \ldots, V_{i-1}$, the node $V_i$ becomes a leaf node and the output $O_G$ is $\kappa_{V_i}$-Lipschitz w.r.t to $V_i$ for all $G \in \mathcal{G}$. In addition, assume that for all $G \in \mathcal{G}$, the node $V$'s value has norm at most $s_V$. Let $\mathrm{pr}(V)$ be all the predecessors of $V$ and $s_{\mathrm{pr}(V)}$ be the list of norm upper bounds of the predecessors of $V$.*

*Then, small covering numbers for all of the local composition rules of $V$ with resolution $\epsilon_V$ would imply small covering number for the family of computational graphs with resolution $\sum_V \epsilon_V \kappa_V$:*

$$\log \mathcal{N}(\sum_{V \in \mathcal{V} \setminus \mathcal{I} \cup \{O\}} \kappa_V \epsilon_V + \epsilon_O, O_{\mathcal{G}}, s_{\mathcal{I}}) \leq \sum_{V \in \mathcal{V} \setminus \mathcal{I}} \log \mathcal{N}(\epsilon_V, \mathfrak{R}_V, s_{\mathrm{pr}(V)}) \qquad (5)$$

In Section A we formalize the notion of "cutting" nodes from the graph. The condition that node $V$'s value has norm at most $s_V$ is a simplification made for expositional purposes; our full Theorem A.3 also applies if $O_G$ collapses to a constant whenever node $V$'s value has norm greater than $s_V$. This allows for the softened indicators $\mathbb{1}_{\leq s_i}(\|h_i(x)\|)$ used in (3).

## 4.2 Lipschitz Augmentation of Computational Graphs

The covering number bound of Theorem 4.3 relies on Lipschitzness w.r.t internal nodes of the graph under a worst-case choice of inputs. For deep networks, this can scale exponentially in depth via the product of weight norms and easily be larger than the average Lipschitz-ness over typical inputs. In this section, we explain a general operation to augment sequential graphs (such as neural nets) into graphs with better worst-case Lipschitz constants, so tools such as Theorem 4.3 can be applied. *This section is heavily simplified for space constraints. Formal definitions and theorem statements are in Section B.*

The augmentation relies on introducing terms such as the soft indicators in equation (2) and (3) which condition on data-dependent properties. As outlined in Section 4, they will translate to the data-dependent properties in the generalization bounds. We also require the augmented function to upper bound the original.

We will present a generic approach to augment function compositions such as $z \triangleq \ell \circ f_k \circ \ldots \circ f_1$, whose Lipschitz constants are potentially exponential in depth, with only properties involving the norms of the inter-layer Jacobians. We will produce $\tilde{z}$, whose worst-case Lipschitzness w.r.t. internal nodes can be polynomial in depth.

**Informal explanation of Lipschitz augmentation:** In the same setting of Section 4, recall that in (2), our first unsuccessful attempt to smooth out the function was by multiplying indicators on the norms of the derivatives of the output: $\prod_{i=1}^{k} \mathbb{1}_{\leq \kappa_i}(\|\partial z / \partial h_i\|_{\mathrm{op}})$. The difficulty lies in controlling the Lipschitzness of the new terms $\|\partial z / \partial h_i\|_{\mathrm{op}}$ that we introduce: by the chain rule, we have the expansion $\frac{\partial z}{\partial h_i} = \frac{\partial z}{\partial h_k} \frac{\partial h_k}{\partial h_{k-1}} \cdots \frac{\partial h_{i+1}}{\partial h_i}$, where each $h_{j'}$ is itself a function of $h_j$ for $j' > j$. This means $\frac{\partial z}{\partial h_i}$ is a complicated function in the intermediate variables $h_j$ for $1 \leq j \leq k$. Bounding the Lipschitzness of $\frac{\partial z}{\partial h_i}$ requires accounting for the Lipschitzness of every term in its expansion, which is challenging and creates complicated dependencies between variables.

Our key insight is that by considering a more complicated augmentation which conditions on the derivatives between all intermediate variables, we can still control Lipschitzness of the system, leading to the more involved augmentation presented in (3). Our main technical contribution is Theorem 4.4, which we informally state below.

**Theorem 4.4** (Informal version of Theorem B.2). *The functions $\tilde{z}$ (defined in (3)) can be computed by a family of computational graphs $\widetilde{\mathcal{G}}$ illustrated in Figure 2. This family has internal nodes $V_i$ and $J_i$ computing $h_i$ and $Df_i[h_{i-1}]$, respectively, and computes a modified output rule that augments the*

*original with soft indicators. These soft indicators condition that the norms of the Jacobians and $h_i$ are bounded by parameters $\kappa_{j\leftarrow i}, s_i$.*

*Importantly, the output $O_{\tilde{G}}$ is $\tilde{\kappa}_{V_i}$, $\tilde{\kappa}_{J_i}$-Lipschitz w.r.t. $V_i$, $J_i$, respectively, after cutting nodes $V_1, J_1, \ldots, V_{i-1}, J_{i-1}$, for parameters $\tilde{\kappa}_{V_i}, \tilde{\kappa}_{J_i}$ that are polynomials in $\kappa_{j\leftarrow i}, s_i$.*

In addition, the augmented function $\tilde{z}$ will upper bound the original with equality when all the indicators are satisfied. The crux of the proof is leveraging the chain rule to decompose $\frac{\partial z}{\partial h_i}$ into a product and then applying a telescoping argument to bound the difference in the product by differences in individual terms. In Section B we present a formal version of this result and also apply Theorem 4.3 to produce a covering number bound for $\widetilde{\mathcal{G}}$.

## 5   Application to Neural Networks

In this section we provide our generalization bound for neural nets, which was obtained using machinery from Section 4.1. Define a neural network $F$ parameterized by $r$ weight matrices $\{W^{(i)}\}$ by $F(x) = W^{(r)}\phi(\cdots\phi(W^{(1)}(x))\cdots)$. We use the convention that activations and matrix multiplications are treated as distinct layers indexed with a subscript, with odd layers applying a matrix multiplication and even layers applying $\phi$ (see Example A.1 for a visualization). Additional notation details and the proof are in Section C.

Figure 2: Lipschitz augmentation (informally defined).

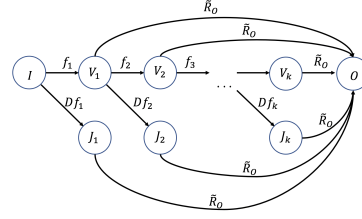

The below result follows from modeling the neural net loss as a sequential computational graph and using our augmentation procedure to make it Lipschitz in its nodes with parameters $\kappa^{\text{hidden},(i)}, \kappa^{\text{jacobian},(i)}$. Then we cover the augmented loss to bound its Rademacher complexity.

**Theorem 5.1.** *Assume that the activation $\phi$ is 1-Lipschitz with a $\bar{\sigma}_\phi$-Lipschitz derivative. Fix reference matrices $\{A^{(i)}\}, \{B^{(i)}\}$. With probability $1-\delta$ over the random draws of the data $P_n$, all neural networks $F$ with parameters $\{W^{(i)}\}$ and positive margin $\gamma$ satisfy:*

$$\mathop{\mathbb{E}}_{(x,y)\sim P}[l_{0\text{-}1}(F(x),y)] \leq \tilde{O}\left(\frac{\left(\sum_i (\kappa^{\text{hidden},(i)}a^{(i)}t^{(i-1)})^{2/3} + (\kappa^{\text{jacobian},(i)}b^{(i)})^{2/3}\right)^{3/2}}{\sqrt{n}} + r\sqrt{\frac{\log(1/\delta)}{n}}\right)$$

*where $\kappa^{\text{jacobian},(i)} \triangleq \sum_{1\leq j\leq 2i-1\leq j'\leq 2r-1} \frac{\sigma_{j'\leftarrow 2i}\sigma_{2i-2\leftarrow j}}{\sigma_{j'\leftarrow j}}$, and $\kappa^{\text{hidden},(i)} \triangleq \xi + \frac{\sigma_{2r-1\leftarrow 2i}}{\gamma} + \sum_{i\leq i'<r}\frac{\sigma_{2i'\leftarrow 2i}}{t^{(i')}} + \sum_{1\leq j\leq j'\leq 2r-1}\sum_{\substack{j''=\max\{2i,j\}, \\ j'' \text{ even}}}^{j'} \frac{\bar{\sigma}_\phi\sigma_{j'\leftarrow j''+1}\sigma_{j''-1\leftarrow 2i}\sigma_{j''-1\leftarrow j}}{\sigma_{j'\leftarrow j}}$.*

*In these expressions, we define $\sigma_{j-1\leftarrow j} = 1$, $\xi = \text{poly}(r)^{-1}$, and:*

$$a^{(i)} \triangleq \|W^{(i)\top} - A^{(i)\top}\|_{2,1} + \xi, b^{(i)} \triangleq \|W^{(i)} - B^{(i)}\|_{1,1} + \xi$$

$$t^{(0)} \triangleq \max_{x\in P_n}\|x\| + \xi, \ t^{(i)} \triangleq \max_{x\in P_n}\|F_{2i-1}(x)\| + \xi$$

$$\sigma_{j'\leftarrow j} \triangleq \max_{x\in P_n}\|Q_{j'\leftarrow j}(x)\|_{\text{op}} + \xi, \text{ and } \gamma \triangleq \min_{(x,y)\in P_n}[F(x)]_y - \max_{y'\neq y}[F(x)]_{y'} > 0$$

*where $Q_{j'\leftarrow j}$ computes the Jacobian of layer $j'$ w.r.t. layer $j$. Note that the training error here is $0$ because of the existence of positive margin $\gamma$.*

We note that our bound has no explicit dependence on width and instead depends on the $\|\cdot\|_{2,1}, \|\cdot\|_{1,1}$ norms of the weights offset by reference matrices $\{A^{(i)}\}, \{B^{(i)}\}$. These norms can avoid scaling with the width of the network if the difference between the weights and reference matrices is sparse. The reference matrices $\{A^{(i)}\}, \{B^{(i)}\}$ are useful if there is some prior belief before training about what weight matrices are learned, and they also appear in the bounds of Bartlett et al. [2017]. In Section G, we also show that our techniques can easily be extended to provide generalization bounds for RNNs scaling polynomially in depth via the same quantities $t^{(i)}, \sigma_{j'\leftarrow j}$.

Table 1: Test error for a model trained on CIFAR10 in various settings.

| Setting | Normalization | Jacobian Reg | Test Error |
|---------|---------------|--------------|------------|
| Baseline | BatchNorm | ✗ | 4.43% |
| Low learning rate (0.01) | BatchNorm | ✗ | 5.98% |
| | | ✓ | **5.46%** |
| No data augmentation | BatchNorm | ✗ | 10.44% |
| | | ✓ | **8.25%** |
| No BatchNorm | None | ✗ | 6.65% |
| | LayerNorm [Ba et al., 2016] | ✗ | 6.20% |
| | | ✓ | **5.57%** |

## 6  Experiments

Though the main purpose of the paper is to study the data-dependent generalization bounds from a theoretical perspective, we provide preliminary experiments demonstrating that the proposed complexity measure and generalization bounds are empirically relevant. We show that regularizing the complexity measure leads to better test accuracy. Inspired by Theorem 5.1, we directly regularize the Jacobian of the classification margin w.r.t outputs of normalization layers and after residual blocks. Our reasoning is that normalization layers control the hidden layer norms, so additionally regularizing the Jacobians results in regularization of the product, which appears in our bound. We find that this is effective for improving test accuracy in a variety of settings. We note that Sokolić et al. [2017] show positive experimental results for a similar regularization technique in data-limited settings.

Suppose that $m(F(x), y) = [F(x)]_y - \max_{j \neq y} [t]_j$ denotes the margin of the network for example $(x, y)$. Letting $h^{(i)}$ denote some hidden layer of the network, we define the notation $J^{(i)} \triangleq \frac{\partial}{\partial h^{(i)}} m(F(x), y)$ and use training objective

$$\hat{L}_{\text{reg}}[F] \triangleq \mathbb{E}_{(x,y) \sim P_n} \left[ l(x, y) + \lambda \left( \sum_i \mathbb{1}(\|J^{(i)}(x)\|_F^2 \geq \sigma) \|J^{(i)}(x)\|_F^2 \right) \right]$$

where $l$ denotes the standard cross entropy loss, and $\lambda, \sigma$ are hyperparameters. Note the Jacobian is taken with respect to a scalar output and therefore is a vector, so it is easy to compute.

For a WideResNet16 [Zagoruyko and Komodakis, 2016] architecture, we train using the above objective. The threshold on the Frobenius norm in the regularization is inspired by the truncations in our augmented loss (in all our experiments, we choose $\sigma = 0.1$). We tune the coefficient $\lambda$ as a hyperparameter. In our experiments, we took the regularized indices $i$ to be last layers in each residual block as well as layers in residual blocks following a BatchNorm in the standard WideResNet16 architecture. In the LayerNorm setting, we simply replaced BatchNorm layers with LayerNorm. The remaining hyperparameter settings are standard for WideResNet; for additional details see Section I.1.

Figure 1 shows the results for models trained and tested on CIFAR10 in low learning rate and no data augmentation settings, which are settings where generalization typically suffers. We also experiment with replacing BatchNorm layers with LayerNorm and additionally regularizing the Jacobian. We observe improvements in test error for all these settings. In Section I.2, we empirically demonstrate that our complexity measure indeed avoids the exponential scaling in depth for a WideResNet model trained on CIFAR10.

## 7  Conclusion

In this paper, we tackle the question of how data-dependent properties affect generalization. We prove tighter generalization bounds that depend polynomially on the hidden layer norms and norms of the interlayer Jacobians. To prove these bounds, we work with the abstraction of computational graphs and develop general tools to augment any sequential family of computational graphs into a Lipschitz family and then cover this Lipschitz family. This augmentation and covering procedure applies to any sequence of function compositions. An interesting direction for future work is to generalize our techniques to arbitrary computational graph structures. Additionally, encouraged by our promising preliminary results, we believe there is the exciting empirical direction of applying these bounds to develop better data-dependent regularization.

## Acknowledgments

CW was supported by a NSF Graduate Research Fellowship. Toyota Research Institute (TRI) provided funds to assist the authors with their research but this article solely reflects the opinions and conclusions of its authors and not TRI or any other Toyota entity.

## Footnotes

[1] A priori, it's also unclear what "Lipschitz in $h_i$" means since the $\bar{z}'$ does not only depend on $x$ through $h_i$. We will formalize this in later section after defining proper language about dependencies between variables.

[2] Unlike in equation (2), we don't augment the Jacobian of the loss w.r.t the layers. This allows us to deal with non-differentiable loss functions such as ramp loss.

[3]As mentioned in footnote 1, we will formalize the precise meaning of Lipschitzness later.

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
