[Supplementary Material]

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

[4]We say the Lipschitzness required is worst case because the release-Lipschitz condition requires the Lipschitzness of nodes for any possible choice of inputs

[5]Note that $DR_{V_i}$ maps a vector in space $\mathcal{D}_{V_{i-1}}$ to an linear operator that maps $\mathcal{D}_{V_{i-1}}$ to $\mathcal{D}_{V_i}$.

[6]Our bound as stated in the paper technically does not apply to ResNet because the skip connections complicate the Lipschitz augmentation step. This can be remedied with a slight modification to our augmentation step, which we omit for simplicity.

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

Figure 3: The computational graph corresponding to a neural network with $r$ weight matrices. Odd-indexed layers multiply matrices and even-indexed layers apply the activation $\phi$.

## A  Covering of Computational Graphs

This section is a formal version of Section 4.1 with full definition and theorem statements. In this section, we adapt the notion of a computational graph to our setting. In Section A.1, we formalize the notion of a computational graph and demonstrate how neural networks fit under this framework. In Section A.2, we define the notion of release-Lipschitzness that abstracts the sequential notion of Lipschitzness in Lemma 4.1. We show that when this release-Lipschitzness condition and a boundedness condition on the internal nodes hold, it is possible to cover a family of computational graphs by simply covering the function class at each vertex.

### A.1  Formalization of computational graphs

When we augment the neural network loss with data-dependent properties, we introduce dependencies between the various layers, making it complicated to cover the augmented loss. We use the notion of computational graphs to abstractly model these dependencies.

Computational graphs are originally introduced by Bauer [1974] to represent computational processes and study error propagation. Recall the notation $G(\mathcal{V}, \mathcal{E}, \{R_V\})$ introduced for a computational graph in Section 4.1, with input nodes $\mathcal{I}_G = \{I_1, \ldots, I_p\}$ and output node denoted by $O_G$. (It's straightforward to generalize to scenarios with multiple output nodes.)

For every variable $V \in \mathcal{V}$, let $\mathcal{D}_V$ be the space that $V$ resides in. If $V$ has $t$ direct predecessors $C_1, \ldots, C_t$, then the associated composition rule $R_V$ is a function that maps $\mathcal{D}_{C_1} \otimes \cdots \otimes \mathcal{D}_{C_t}$ to $\mathcal{D}_V$. If $V$ is an input node, then the composition rule $R_V$ is not relevant. For any node $V$, the computational graph defines/induces a function that computes the variable $V$ from inputs, or in mathematical words, that maps the inputs space $\mathcal{D}_{I_1} \otimes \cdots \otimes \mathcal{D}_{I_p}$ to $\mathcal{D}_V$. This associated function, denoted by $V$ again with slight abuse of notations, is defined recursively as follows: set $V(x_1, \ldots, x_p)$ to

$$\begin{cases} x_i & \text{if } V \text{ is the } i\text{-th input node } I_i \\ R_V(C_1(x_1, \ldots, x_p), \ldots, C_t(x_1, \ldots, x_p)) & \text{if } V \text{ has } t \text{ direct predecessors } C_1, \ldots, C_t \end{cases}$$

More succinctly, we can write $V = R_V \circ (C_1 \otimes \cdots \otimes C_t)$. We also overload the notation $O_G$ to denote the function that the computational graph $G$ finally computes (which maps $\mathcal{D}_{I_1} \otimes \cdots \otimes \mathcal{D}_{I_p}$ to $\mathcal{D}_O$). For any set $\mathcal{S} = \{V_1, \ldots, V_t\} \subseteq \mathcal{V}$, use $\mathcal{D}_\mathcal{S}$ to denote the space $\mathcal{D}_{V_1} \otimes \cdots \otimes \mathcal{D}_{V_t}$. We use $\mathrm{pr}(G, V)$ to denote the set of direct predecessors of $V$ in graph $G$, or simply $\mathrm{pr}(V)$ when the graph $G$ is clear from context.

**Example A.1** (Feed-forward neural networks). *For an activation function $\phi$ and parameters $\{W^{(i)}\}$ we compute a neural net $F : \mathbb{R}^{d_I} \to \mathbb{R}^{d_O}$ as follows: $F(x) = W^{(r)}\phi(\cdots\phi(W^{(1)}x)\cdots)$. Figure 3 depicts how this neural network fits into a computational graph with one input node, $2r - 1$ internal nodes, and a single output. Here we treat matrix operations and activations as distinct layers, and map each layer to a node in the computational graph.*

### A.2  Reducing graph covering to local function covering

In this section we introduce the notion of a family of computational graphs, generalizing the sequential family of function compositions in (1). We define release-Lipschitzness, a condition which allows reduce covering the entire the graph family to covering the composition rules at each node. We formally state this reduction in Theorem A.3.

**Family of computational graphs:**  Let $\mathcal{G} = \{G(\mathcal{V}, \mathcal{E}, \{R_V\}) : \{R_V\} \in \mathfrak{R}\}$ be a family of computational graph with shared nodes and edges, where $\mathfrak{R}$ is a collection of lists of composition rules. This family of computational graphs defines a set of functions $O_\mathcal{G} \triangleq \{O_G : G \in \mathcal{G}\}$. We'd like to cover this set of functions in $O_\mathcal{G}$ with respect to some metric $L(P_n, \|\|\cdot\|\|)$.

For a list of composition rules $\{R_V\} \in \mathfrak{R}$ and subset $\mathcal{S} \subseteq \mathcal{V}$, we define the projection of composition rules onto $\mathcal{S}$ by $\{R_V\}_\mathcal{S} = \{R_V : V \in \mathcal{S}\}$. Now let $\mathfrak{R}_\mathcal{S} = \{\{R_V\}_\mathcal{S} : \{R_V\} \in \mathfrak{R}\}$ denote the marginal collection of the composition rules on node subset $\mathcal{S}$.

For any computational graph $G$ and a non-input node $V \in \mathcal{V} \setminus \mathcal{I}$, we can define the following operation that "releases" $V$ from its dependencies on its predecessors by cutting all the inward edges: Let $G^{\backslash V}$ be sub-graph of $G$ where all the edges pointing towards $V$ are removed from the graph. Thus, by definition, $V$ becomes a new input node of the graph $G^{\backslash V}$: $\mathcal{I}_{G^{\backslash V}} = \{V\} \cup \mathcal{I}_G$. Moreover, we can "recover" the dependency by plugging the right value for $V$ in the new graph $G^{\backslash V}$: Let $V(x)$ be the function associated to the node $V$ in graph $G$, then we have

$$\forall x \in \mathcal{D}_\mathcal{I}, \ O_{G^{\backslash V}}(V(x), x) = O_G(x) \tag{6}$$

In our proofs, we will release variables in orders. Let $\mathcal{S} = (V_1, \ldots, V_m)$ be an ordering of the intermediate variables $\mathcal{V} \setminus (\mathcal{I} \cup \{O\})$. We call $\mathcal{S}$ a forest ordering if for any $i$, in the original graph $G$, $V_i$ at most depends on the input nodes and $V_1, \ldots, V_{i-1}$. For any sequence of variables $(V_1, \ldots, V_t)$, we can define the graph obtained by releasing the variables in order: $G^{\backslash(V_1, \ldots, V_t)} \triangleq (\cdots (G^{\backslash V_1}) \cdots)^{\backslash V_t}$. We next define the release-Lipschitz condition, which states that the graph function remains Lipschitz when we sequentially release vertices in a forest ordering of the graph.

**Definition A.2** (Release-Lipschitzness). *A graph $G$ is release-Lipschitz with parameters $\{\kappa_V\}$ w.r.t a forest ordering of the internal nodes, denoted by $(V_1, \ldots, V_m)$ if the following happens: upon releasing $V_1, \ldots, V_m$ in order from any $G \in \mathcal{G}$, for any $0 \leq i \leq m$, we have that the function defined by the released graph $G^{\backslash(V_1, \ldots, V_i)}$ is $\kappa_{V_i}$-Lipschitz in the argument $V_i$, for any values of the rest of the input nodes ($= \{V_1, \ldots, V_{i-1}\} \cup \mathcal{I}_G$.) We also say graph $G$ is release-Lipschitz if such a forest ordering exists.*

Now we show that the release-Lipschitz condition allows us to cover any family of computational graphs whose output collapses when internal nodes are too large. The below is a formal and complete version of Theorem 4.3. For the augmented loss defined in (3), the function output collapses to 1 when internal computations are large. The proof is deferred to Section D.

**Theorem A.3.** *Suppose $\mathcal{G}$ is a computational graph with the associated family of lists of composition rules $\mathfrak{R}$, as formally defined above. Let $P_n$ be a uniform distribution over $n$ points in $\mathcal{D}_\mathcal{I}$. Let $\kappa_V, s_V,$ and $\epsilon_V$ be three families of fixed parameters indexed by $\mathcal{V} \setminus \mathcal{I}$ (whose meanings are defined below). Assume the following:*

1. *Every $G \in \mathcal{G}$ is release-Lipschitz with parameters $\{\kappa_V\}$ w.r.t a forest ordering of the internal nodes $(V_1, \ldots, V_m)$ (the parameter $\kappa_V$'s and ordering doesn't depend on the choice of $G$.)*

2. *For the same order as before, if $(v, x) \in (\mathcal{D}_{V_1} \otimes \cdots \otimes \mathcal{D}_{V_i}) \otimes \mathcal{D}_\mathcal{I}$ is an input of the released graph satisfying $\|v_j\| \geq s_{V_j}$ for some $j \leq i$, then $O_{G^{\backslash(V_1, \ldots, V_i)}}(v, x) = c$ for some constant $c$.*

*Then, small covering numbers for all of the local composition rules of $V$ with resolution $\epsilon_V$ would imply small covering number for the family of computational graphs with resolution $\sum_V \epsilon_V \kappa_V$:*

$$\log \mathcal{N}(\sum_{V \in \mathcal{V} \setminus \mathcal{I} \cup \{O\}} \kappa_V \epsilon_V + \epsilon_O, O_\mathcal{G}, s_\mathcal{I}) \leq \sum_{V \in \mathcal{V} \setminus \mathcal{I}} \log \mathcal{N}(\epsilon_V, \mathfrak{R}_{\{V\}}, s_{\text{pr}(V)}) \tag{7}$$

# B  Lipschitz Augmentation of Computational Graphs

In this section, we provide a more thorough and formal presentation of the augmentation framework of Section 4.2.

The covering number bound for the computational graph family $\mathcal{G}$ in Theorem A.3 relies on the release-Lipschitzness condition (condition 1 of Theorem A.3) and rarely holds for deep computational graphs such as deep neural networks. The conundrum is that the worst-case Lipschitzness as required in the release-Lipschitz condition[4] is very likely to scale in the product of the worst-case Lipschitzness of

each operations in the graph, which can easily be exponentially larger than the average Lipschitzness over typical examples.

In this section, we first define a model of sequential computational graphs, which captures the class of neural networks. Before Lipschitz augmentation, the worst-case Lipschitz constant of graphs in this family could scale exponentially in the depth of the graph. In Definition B.1, we generalize the operation of (3) to augment any family $\mathcal{G}$ of sequential graphs and produce a family $\widetilde{\mathcal{G}}$ satisfying the release-Lipschitz condition. In Theorem B.3, we combine this augmentation with the framework of A.3 to produce general covering number bounds for the augmented graphs. For the rest of this section we will work with sequential families of computational graphs.

A sequential computational graph has nodes set $\mathcal{V} = \{I, V_1, \ldots, V_q, O\}$, where $I$ is the single input node, and all the edges are $\mathcal{E} = \{(I, V_1), (V_1, V_2), \cdots, (V_{q-1}, V_q)\} \cup \{(V_1, O), \ldots, (V_q, O)\}$. We often use the notation $V_0$ to refer to the input $I$. Below we formally define the augmentation operation.

**Definition B.1** (Lipschitz augmentation of sequential graphs). *Given a differentiable sequential computational graph $G$ with $q$ internal nodes $V_1, \ldots, V_q$, define its Lipschitz augmentation $\widetilde{G}$ as follows. We first add $q$ nodes to the graph denoted by $J_1, \ldots, J_q$. The composition rules for original internal nodes remain the same, and the composition rule for $J_i$ is defined as*

$$\widetilde{R}_{J_i} = DR_{V_i}$$

*Here $DR_{V_i}$ is the total derivative of the function $R_{V_i}$. In other words, the variable $J_i$ is a Jacobian for $R_{V_i}$, a linear operator that maps $\mathcal{D}_{V_{i-1}}$ to $\mathcal{D}_{V_i}$. (Note that if $V_i$'s are considered as vector variables, then $J_i$'s are matrix variables.) We equip the space of $J_i$ with operator norm, denoted by $\|\cdot\|_{\mathrm{op}}$, induced by the original norms on spaces $V_{i-1}$ and $V_i$. The Lipschitz-ness w.r.t variable $J_i$ will be measured with operator norm.*

*We pre-determine a family of parameters $\kappa_{j \leftarrow i}$ for all pairs $(i, j)$ with $i \leq j$. The final loss is augmented by a product of soft indicators that truncates the function when any of the Jacobians is much larger than $\kappa_{i \leftarrow j}$:*

$$\widetilde{R}_O(x, v_1, \ldots, v_q, D_1, \ldots, D_q) \triangleq (R_O(x, v_1, \ldots, v_q) - 1) \prod_{i \leq j} \mathbb{1}_{\leq \kappa_{j \leftarrow i}}(\|D_j \cdots D_i\|_{\mathrm{op}}) + 1$$

*where $x \in \mathcal{D}_\mathcal{I}$, $v_i \in \mathcal{D}_{V_i}$, and $D_i \in \mathcal{D}_{J_i}$. Note that $D_j \cdots D_i$ is the total derivative of $V_j$ w.r.t $V_i$, and thus the $\kappa_{j \leftarrow i}$ has the interpretation as an intended bound of the Jacobian between pairs of layers (variables). Figure 4 depicts the augmentation.*

Note that under these definitions, we finally get that the output function of $\widetilde{G}$ computes

$$O_{\widetilde{G}}(x) = (O_G(x) - 1) \prod_{i \leq j} \mathbb{1}_{\leq \kappa_{j \leftarrow i}}(\|DV_j(x) \cdots DV_i(x)\|_{\mathrm{op}}) + 1 \tag{8}$$

which matches (3) for the example in Section 4. We note that the graph $\tilde{G}$ contains the original $G$ as a subgraph. Furthermore, by Claim J.1, $O_{\widetilde{G}}$ upper bounds $O_G$, which is desirable when $G$ computes loss functions. The below theorem, which formalizes Theorem 4.4, proves release-Lipschitzness for $\widetilde{\mathcal{G}}$.

**Theorem B.2.** *[Lipschitz guarantees of augmented graphs] Let $\mathcal{G}$ be a family of sequential computational graphs. Suppose for any $G \in \mathcal{G}$, the composition rule of the output node, $R_{O_G}$, is $c_i$-Lipschitz in variable $V_i$ for all $i$, and it only outputs value in $[0, 1]$. Suppose that $DR_{V_i}$ is $\bar{\kappa}_i$-Lipschitz for each $i$.[5] Let $\kappa_{j \leftarrow i}$ (for $i \leq j$) be a set of parameters that we intend to use to control Jacobians in the Lipschitz augmentation. With them, we apply Lipschitz augmentation as defined in Definition B.1 to every graph in $\mathcal{G}$ and obtain a new family of graphs, denoted by $\widetilde{\mathcal{G}}$.*

*Then, the augmented family $\widetilde{\mathcal{G}}$ is release-Lipschitz (Definition A.2) with parameters $\tilde{\kappa}_V$'s below:*

$$\tilde{\kappa}_{V_i} \triangleq \sum_{i \leq j \leq q} 3c_j \kappa_{j \leftarrow i+1} + 18 \sum_{1 \leq j \leq j' \leq q} \sum_{i'=\max\{i+1, j\}}^{j'} \frac{\bar{\kappa}_{i'} \kappa_{j' \leftarrow i'+1} \kappa_{i'-1 \leftarrow i+1} \kappa_{i'-1 \leftarrow j}}{\kappa_{j' \leftarrow j}},$$

$$\tilde{\kappa}_{J_i} \triangleq \sum_{j \leq i \leq j'} \frac{4\kappa_{j' \leftarrow i+1} \kappa_{i-1 \leftarrow j}}{\kappa_{j' \leftarrow j}}$$

*where for simplicity in the above expressions, we extend the definition of $\kappa$'s to $\kappa_{j-1 \leftarrow j} = 1$.*

Finally, we combine Theorems A.3 and Theorems B.2 to derive covering number bounds for any Lipschitz augmentation of sequential computational graphs. The final covering bound in (9) can be easily computed given covering number bounds for each individual function class. In Section 5, we use this theorem to derive Rademacher complexity bounds for neural networks. The proof is deferred to Section E. In Section G, we also use these tools to derive Rademacher complexity bounds for RNNs.

Figure 4: Lipschitz augmentation (formally defined).

**Theorem B.3.** *Consider any family $\mathcal{G}$ of sequential computational graphs satisfying the conditions of Theorem B.2. By combining the augmentation of Definition B.1 with additional indicators on the internal node norms, we can construct a new family $\widetilde{\mathcal{G}}$ of computational graphs which output*

$$O_{\widetilde{G}}(x) = (O_G(x) - 1) \prod_{i=1}^{q} \mathbb{1}_{\leq s_{V_i}}(\|V_i(x)\|) \prod_{1 \leq i \leq j \leq q} \mathbb{1}_{\leq \kappa_{j \leftarrow i}}(\|DV_j(x) \cdots DV_i(x)\|_{\mathrm{op}}) + 1$$

*The family $\widetilde{\mathcal{G}}$ satisfies the following guarantees:*

1. *Each computational graph in $\widetilde{\mathcal{G}}$ upper bounds its counterpart in $\mathcal{G}$, i.e. $O_{\widetilde{G}}(x) \geq O_G(x)$.*

2. *Define $\tilde{\kappa}'_{V_i} \triangleq \tilde{\kappa}_{V_i} + \sum_{i \leq j \leq q} s_{V_j}^{-1} \cdot \kappa_{j \leftarrow i+1}$ and $\tilde{\kappa}'_{J_i} = \tilde{\kappa}_{J_i}$ where $\tilde{\kappa}_{V_i}, \tilde{\kappa}_{J_i}$ are defined as in Theorem B.2. Then for any node-wise errors $\{\epsilon_V\}$,*

$$\log \mathcal{N}(\sum_{i \geq 1} \tilde{\kappa}'_{V_i} \epsilon_{V_i} + \tilde{\kappa}_{J_i} \epsilon_{J_i} + \epsilon_O, O_{\widetilde{\mathcal{G}}}, s_{\mathcal{I}}) \tag{9}$$

$$\leq \sum_{i \geq 1} \log \mathcal{N}(\epsilon_{V_i}, \mathfrak{R}_{V_i}, 2s_{V_{i-1}}) + \log \mathcal{N}(\epsilon_{J_i}, D\mathfrak{R}_{V_i}, 2s_{V_{i-1}}) + \log \mathcal{N}(\epsilon_O, \mathfrak{R}_O, \{2s_{V_j}\}_{j=1}^{q} \cup \{I\})$$

*where $D\mathfrak{R}_{V_i}$ denotes the family of total derivatives of functions in $\mathfrak{R}_{V_i}$ and $V_0$ the input vertex.*

# C  Missing Proofs for Section 5

We first elaborate more on the notations introduced in Section 5. First, by our indexing, matrix $W^{(i)}$ will be applied in layer $2i - 1$ of the network, and even layers $2i$ apply $\phi$. We let $F_{j' \leftarrow j}$ denote the function computed between layers $j$ and $j'$ and $Q_{j' \leftarrow j} = DF_{j' \leftarrow j} \circ F_{j'-1 \leftarrow 1}$ denote the layer $j$-to-$j'$ Jacobian. By our definition of $F_{j' \leftarrow j}$, $F_{2j \leftarrow 2j} = \phi$, $F_{2j-1 \leftarrow 2j-1} = h \mapsto W^{(j)} h$, and $F_{j' \leftarrow j}$ is recursively computed by $F_{j' \leftarrow j'} \circ F_{j'-1 \leftarrow j}$ for $j' > j$. We will use the convention that $F_{j-1 \leftarrow j}$ computes the identity mapping for $i \leq j$.

$P$ will denote a test distribution over examples $x$ and labels $y$, and $P_n$ will denote the distribution on training examples.

For a class of real-valued functions $\mathcal{L}$ and dataset $P_n$, define the empirical Rademacher complexity of this function class by

$$\mathrm{Rad}_n(\mathcal{L}) = \frac{1}{n} \mathbb{E}_{\alpha_i} \left[ \sup_{l \in \mathcal{L}} \sum_i \alpha_i l(x_i) \right] \tag{10}$$

where $\alpha_i$ are independent uniform $\pm 1$ random variables. Let $m(t, y) \triangleq [t]_y - \max_{j \neq y} [t]_j$ denote the margin operator for label $y$, and $l_\gamma(t, y) \triangleq \mathbb{1}(m(t, y) \leq 0) - \mathbb{1}(0 < m(t, y) \leq \gamma) \cdot m(t, y)/\gamma$ denote the standard ramp loss, which is $1/\gamma$-Lipschitz. We will work in the neural network setting defined in Section 5. We will first state our generalization bound for neural networks.

**Theorem C.1.** *Assume that the activation $\phi$ is 1-Lipschitz with $\bar{\sigma}_\phi$-Lipschitz derivative. Fix parameters $\sigma_{j' \leftarrow j}, t^{(i)}, a^{(i)}, b^{(i)}, \gamma$ and reference matrices $\{A^{(i)}\}, \{B^{(i)}\}$. With probability $1 - \delta$ over the random draws of the distribution $P_n$, all neural networks $F$ with parameters $\{W^{(i)}\}$ satisfying the following data-dependent conditions:*

1. *Hidden layers norms are controlled:* $\max_{x \in P_n} \|F_{2i \leftarrow 1}(x)\| \leq t^{(i)} \ \forall 1 \leq i \leq r$.

2. *Jacobians are balanced:* $\max_{x \in P_n} \|Q_{j' \leftarrow j}(x)\|_{\text{op}} \leq \sigma_{j' \leftarrow j} \ \forall j < j'$.

3. *The margin is large:* $\min_{(x,y) \in P_n} [F(x)]_y - \max_{y' \neq y} [F(x)]_{y'} \geq \gamma > 0$.

*and the additional data-independent condition*

$$\|W^{(i)^\top} - A^{(i)^\top}\|_{2,1} \leq a^{(i)}, \|W^{(i)} - B^{(i)}\|_{1,1} \leq b^{(i)}, \|W^{(i)}\|_{\text{op}} \leq \sigma_{2i-1 \leftarrow 2i-1}$$

*will have the following generalization to test data:*

$$\underset{(x,y) \sim \mathcal{D}}{\mathbb{E}}[l_{\text{0-1}}(F(x), y)] \leq \tilde{O}\left(\frac{\left(\sum_i (\kappa^{\text{hidden},(i)} a^{(i)} t^{(i-1)})^{2/3} + (\kappa^{\text{jacobian},(i)} b^{(i)})^{2/3}\right)^{3/2}}{\sqrt{n}}\right) + \sqrt{\frac{\log(1/\delta)}{n}}$$

*where*

$$\kappa^{\text{jacobian},(i)} \triangleq \sum_{1 \leq j \leq 2i-1 \leq j' \leq 2r-1} \frac{4\sigma_{j' \leftarrow 2i} \sigma_{2i-2 \leftarrow j}}{\sigma_{j' \leftarrow j}} \tag{11}$$

$$\kappa^{\text{hidden},(i)} \triangleq \frac{\sigma_{2r-1 \leftarrow 2i}}{\gamma} + \sum_{i \leq i' < r} \frac{3\sigma_{2i' \leftarrow 2i}}{t^{(i')}}$$

$$+ \sum_{1 \leq j \leq j' \leq 2r-1} \sum_{\substack{j'' = \max\{2i,j\}, \\ j'' \text{ even}}}^{j'} \frac{\bar{\sigma}_\phi \sigma_{j' \leftarrow j''+1} \sigma_{j''-1 \leftarrow 2i} \sigma_{j''-1 \leftarrow j}}{\sigma_{j' \leftarrow j}} \tag{12}$$

*Here we use the convention that $\sigma_{j-1 \leftarrow j} = 1$ and let $t^{(0)} = \max_{x \in P_n} \|x\|$.*

This generalization bound follows straightforwardly via the below Rademacher complexity bound for the augmented loss class:

**Theorem C.2.** *Suppose that $\phi$ is 1-Lipschitz with $\bar{\sigma}_\phi$-Lipschitz derivative. Define the following class of neural networks with norm bounds on its weight matrices with respect to reference matrices $\{A^{(i)}\}, \{B^{(i)}\}$:*

$$\mathcal{F} \triangleq \left\{x \mapsto F(x) : \|W^{(i)^\top} - A^{(i)^\top}\|_{2,1} \leq a^{(i)}, \|W^{(i)} - B^{(i)}\|_{1,1} \leq b^{(i)}, \|W^{(i)}\|_{\text{op}} \leq \sigma^{(i)}\right\}$$

*Fix parameters $t^{(i)}$ and $\sigma_{j' \leftarrow j}$ for $j' \geq j$ with $\sigma_{2i \leftarrow 2i} = 1$ and $\sigma_{2i-1 \leftarrow 2i-1} = \sigma^{(i)}$. When we apply this theorem, we will choose $\sigma_{j' \leftarrow j}$ and $t^{(i)}$ which upper bound the layer $j$ to $j'$ Jacobian norm and $i$-th hidden layer norm, respectively. Define the class of augmented losses*

$$\mathcal{L}_{\text{aug}} \triangleq \left\{(l_\gamma - 1) \circ F \prod_{i=1}^{r-1} \mathbb{1}_{\leq t^{(i)}}(\|F_{2i \leftarrow 1}\|) \prod_{1 \leq j < j' \leq 2r-1} \mathbb{1}_{\leq \sigma_{j' \leftarrow j}}(\|Q_{j' \leftarrow j}\|_{\text{op}}) + 1 : F \in \mathcal{F}\right\}$$

*and define for $1 \leq i \leq r$, $\kappa^{\text{jacobian},(i)}, \kappa^{\text{hidden},(i)}$ meant to bound the influence of the matrix $W^{(i)}$ on the Jacobians and hidden variables, respectively as in (11), (12). Then we can bound the empirical Rademacher complexity of the augmented loss class by*

$$\text{Rad}_n(\mathcal{L}_{\text{aug}}) = \tilde{O}\left(\frac{\left(\sum_i (\kappa^{\text{hidden},(i)} a^{(i)} t^{(i-1)})^{2/3} + (\kappa^{\text{jacobian},(i)} b^{(i)})^{2/3}\right)^{3/2}}{\sqrt{n}}\right)$$

*where we recall that the notation $\tilde{O}$ hides log factors in the arguments and the dimension of the weight matrices.*

*Proof.* We associate the un-augmented loss class on neural networks $l_\gamma \circ \mathcal{F}$ with a family of sequential computation graphs $\mathcal{G}$ with depth $2r - 1$. The composition rules are as follows: for internal node $V_{2i}$, $\mathfrak{R}_{V_{2i}} = \{\phi\}$, the set with only one element: the activation $\phi$. We also let $\mathfrak{R}_{V_{2i-1}} = \{h \mapsto Wh : \|W^\top - A^{(i)^\top}\|_{2,1} \leq a^{(i)}, \|W - B^{(i)}\|_{1,1} \leq b^{(i)}, \|W\|_{\text{op}} \leq \sigma^{(i)}\}$. Finally, we

choose $\mathfrak{R}_O$ to be the singleton class $\{l_\gamma\}$. Our collection of computation rules is then simply $\mathfrak{R} = \mathfrak{R}_{V_1} \otimes \cdots \otimes \mathfrak{R}_{V_{2r-1}} \otimes \mathfrak{R}_O$. Since $O_{\mathcal{G}}$ takes values in $[0, 1]$, we can apply Theorem B.3 on this class $\mathcal{G}$ using $s_{\mathcal{I}} = \max_{x \in P_n} \|x\|$, $s_{V_{2i}} = t^{(i)}$, $s_{V_{2i-1}} = \infty$, $\kappa_{2i \leftarrow 2i} = 1$, $\kappa_{2i-1 \leftarrow 2i-1} = \sigma^{(i)}$, and $\kappa_{j' \leftarrow j} = \sigma_{j' \leftarrow j}$ for $j' > j$. Furthermore, we note that $\bar{\kappa}_{2i} = \bar{\sigma}_\phi$, and $\bar{\kappa}_{2i-1} = 0$ as the Jacobian is constant for matrix multiplications. We thus obtain the class $\widetilde{\mathcal{G}}$ where each augmented loss upper bounds the corresponding loss in $\mathcal{G}$. Recall that $J_i$ denote the additional nodes in our augmented computation graph. Note that under these choices of $s_{V_{2i-1}}$, $\kappa_{i \leftarrow i}$, we get that

$$\mathbb{1}_{\leq \kappa_{2i \leftarrow 2i}}(\|J_{2i}(x)\|_{\mathrm{op}}) = \mathbb{1}_{\leq 1}(\|D\phi \circ V_{2i-1}(x)\|_{\mathrm{op}}) = 1 \qquad \text{(as } |\phi'| \leq 1\text{)}$$

$$\mathbb{1}_{\leq \kappa_{2i-1 \leftarrow 2i-1}}(\|J_{2i-1}(x)\|_{\mathrm{op}}) = \mathbb{1}_{\leq \sigma^{(i)}}(\|W^{(i)} \circ V_{2i-2}(x)\|_{\mathrm{op}}) = 1 \qquad \text{(as } W^{(i)} \leq \sigma^{(i)}\text{)}$$

$$\mathbb{1}_{\leq s_{V_{2i-1}}}(\|V_{2i-1}(x)\|) = \mathbb{1}_{\leq \infty}(\|V_{2i-1}(x)\|) = 1$$

Furthermore, the other indicators in the augmented loss map to indicators in the outputs of our augmented graphs $O_{\widetilde{G}}$, so therefore the families $\mathcal{L}_{\mathrm{aug}}$ defined in the theorem statement and $\widetilde{\mathcal{G}}$ are equivalent. Thus, it suffices to bound the Rademacher complexity of $\widetilde{\mathcal{G}}$. To do this, we invoke covering numbers. By Theorem B.3, we bound the covering number of $O_{\widetilde{\mathcal{G}}}$:

$$\log \mathcal{N}(\sum_{i \geq 1}(\tilde{\kappa}_{V_i} + \tilde{\kappa}_{J_i})\epsilon_V + \epsilon_O, O_{\widetilde{\mathcal{G}}}, s_{\mathcal{I}}) \leq$$

$$\sum_{i \geq 1} \log \mathcal{N}(\epsilon_{V_i}, \mathfrak{R}_{V_i}, 2s_{V_{i-1}}) + \log \mathcal{N}(\epsilon_{J_i}, D\mathfrak{R}_{V_i}, 2s_{V_{i-1}}) + \log \mathcal{N}(\epsilon_O, \mathfrak{R}_O, \{2s_{V_i}\}_{i \geq 0}) \tag{13}$$

where $\tilde{\kappa}_{V_i}, \tilde{\kappa}_{J_i}$ are defined in the statement of Theorem B.3. After plugging in our values for $\bar{\kappa}_j$, $s_{V_j}$, $\kappa_{j' \leftarrow j}$ in our application of Theorem B.3 and noting that $c_{2i} = 1/t^{(i)}$, $c_{2i-1} = 0$ for $i < r$ and $1/\gamma$ for $i = r$ (as the margin loss is $1/\gamma$-Lipschitz), we obtain that

$$\tilde{\kappa}_{V_{2i-1}} = \kappa^{\mathrm{hidden},(i)}, \tilde{\kappa}_{J_{2i-1}} = \kappa^{\mathrm{jacobian},(i)}$$

We first note that the last term in (13) is simply 0 because there is exactly one output function in $\mathfrak{R}_O$. Now for the other terms of (13): by definition $\mathfrak{R}_{V_{2i}}, \mathfrak{R}_{J_{2i}}$ consist of a singleton set and therefore have log cover size 0 for any error resolution $\epsilon$. Otherwise, to cover $\mathfrak{R}_{V_{2i-1}}$ it suffices to bound $\log \mathcal{N}(\epsilon_{V_{2i-1}}, \{h \mapsto Wh : \|W^\top - {A^{(i)}}^\top\|_{2,1} \leq a^{(i)}\}, 2t^{(i-1)})$. Thus, we can apply Lemma C.3 to obtain

$$\log \mathcal{N}(\epsilon_{V_{2i-1}}, \mathfrak{R}_{V_{2i-1}}, 2s_{V_{2i-2}}) \leq \tilde{O}\left(\frac{(a^{(i)}t^{(i-1)})^2}{\epsilon_{V_{2i-1}}^2}\right)$$

Now to cover $D\mathfrak{R}_{V_{2i-1}}$, it suffices to cover $\{W : \|W - B^{(i)}\|_{1,1} \leq b^{(i)}\}$. The $\epsilon$-covering number of a $d_h^2$-dimensional $\ell_1$-ball with radius $b$ w.r.t. $\ell_2$ norm is $O(\frac{b^2}{\epsilon^2} \log d_h)$. Thus,

$$\log \mathcal{N}(\epsilon_{J_{2i-1}}, D\mathfrak{R}_{V_{2i-1}}, 2s_{V_{2i-2}}) \leq \tilde{O}\left(\frac{(b^{(i)})^2}{\epsilon_{J_{2i-1}}^2}\right)$$

Now we define

$$\beta^\star \triangleq \left(\sum_i (\tilde{\kappa}_{V_{2i-1}} a^{(i)} t^{(i-1)})^{2/3} + (\tilde{\kappa}_{J_{2i-1}} b^{(i)})^{2/3}\right)^{3/2}$$

$$= \left(\sum_i (\kappa^{\mathrm{hidden},(i)} a^{(i)} t^{(i-1)})^{2/3} + (\kappa^{\mathrm{jacobian},(i)} b^{(i)})^{2/3}\right)^{3/2}$$

Now for a fixed error parameter $\epsilon$, we set $\epsilon_O = 0$, $\epsilon_{V_{2i}} = 0$, $\epsilon_{J_{2i}} = 0$ (as the log cover size is 0 anyways), and $\epsilon_{V_{2i-1}} = \epsilon \frac{\tilde{\kappa}_{V_{2i-1}}^{-1/3}(a^{(i)}t^{(i-1)})^{2/3}}{(\beta^\star)^{2/3}}$, $\epsilon_{J_{2i-1}} = \epsilon \frac{\tilde{\kappa}_{J_{2i-1}}^{-1/3}(b^{(i)})^{2/3}}{(\beta^\star)^{2/3}}$ Now it follows that $\sum_j \epsilon_{V_j}\tilde{\kappa}_{V_i} + \epsilon_{J_j}\tilde{\kappa}_{J_j} = \epsilon$. Furthermore, under these choices of $\epsilon_{V_i}, \epsilon_{J_i}$, we end up with

$$\sum_{i \geq 1} \log \mathcal{N}(\epsilon_{V_i}, \mathfrak{R}_{V_i}, 2s_{V_{i-1}}) + \log \mathcal{N}(\epsilon_{J_i}, D\mathfrak{R}_{V_i}, 2s_{V_{i-1}})$$

$$\leq \tilde{O}\left(\frac{1}{\epsilon^2}(\beta^\star)^{4/3}\left(\sum_i (\kappa^{\mathrm{hidden},(i)} a^{(i)} t^{(i-1)})^{2/3} + (\kappa^{\mathrm{jacobian},(i)} b^{(i)})^{2/3}\right)^{3/2}\right) = \tilde{O}(\epsilon^{-2}(\beta^\star)^2)$$

Thus, substituting terms into (13) and collecting sums, we obtain that

$$\log \mathcal{N}(\epsilon, O_{\widetilde{\mathcal{G}}}, s_{\mathcal{I}}) \leq \tilde{O}(\epsilon^{-2}(\beta^{\star})^2)$$

Now we apply Dudley's entropy theorem to obtain that

$$\text{Rad}_n(\widetilde{\mathcal{G}}) = \tilde{O}\left( \frac{\left(\sum_i (\kappa^{\text{hidden},(i)} a^{(i)} t^{(i-1)})^{2/3} + (\kappa^{\text{jacobian},(i)} b^{(i)})^{2/3}\right)^{3/2}}{\sqrt{n}} \right)$$

$\square$

We now apply C.2 to prove Theorem C.1.

*Proof of Theorem C.1.* We start with Theorem C.2, which bounds the Rademacher complexity of the augmented loss class $\mathcal{L}_{\text{aug}}$. Using $l_{\text{aug}}(F, x, y)$ to denote the application of this augmented loss on the network $F$, its weights, and data $(x, y)$, we first note that $l_{\text{0-1}}(F(x), y) \leq l_{\gamma}(F(x), y) \leq l_{\text{aug}}(F, x, y)$ for any datapoint $(x, y)$. We used the fact that margin loss upper bounds 0-1 loss, and $l_{\text{aug}}$ upper bounds margin loss by the construction in Theorem B.3. Thus, applying the standard Rademacher generalization bound, with probability $1 - \delta$ over the training data, it holds that

$$\mathop{\mathbb{E}}_{(x,y)\sim\mathcal{D}}[l_{\text{0-1}}(F(x), y)] \leq \mathop{\mathbb{E}}_{(x,y)\sim\mathcal{D}}[l_{\text{aug}}(F, x, y)] \tag{14}$$

$$\leq \mathop{\mathbb{E}}_{(x,y)\sim\mathcal{D}_n}[l_{\text{aug}}(F, x, y)] + \text{Rad}_n(\mathcal{L}_{\text{aug}}) + \sqrt{\frac{\log(1/\delta)}{n}} \tag{15}$$

$$= \text{Rad}_n(\mathcal{L}_{\text{aug}}) + \sqrt{\frac{\log(1/\delta)}{n}} \qquad \text{(by the data-dependent conditions)}$$

Plugging in the bound on $\text{Rad}_n(\mathcal{L}_{\text{aug}})$ from Theorem C.2 gives the desired result. $\square$

Finally, to prove Theorems 5.1 and 1.1, we simply take a union bound over the choices of parameters $\sigma_{j'\leftarrow j}, t^{(i)}, a^{(i)}, b^{(i)}$.

*Proof of Theorems 5.1 and 1.1.* We will apply Theorem C.1 repeatedly over a grid of parameter choices $t^{(i)}, \sigma_{j'\leftarrow j}, a^{(i)}, b^{(i)}$ (following a technique of Bartlett et al. [2017]). For a collection $\mathcal{M}$ of nonnegative integers $m_t^{(i)}, m_\sigma^{(j'\leftarrow j)}, m_a^{(i)}, m_b^{(i)}, m_\gamma$, we apply Theorem C.1 choosing $t^{(i)} = \text{poly}(r)^{-1} 2^{m_t^{(i)}}$, $\sigma_{j'\leftarrow j} = \text{poly}(r)^{-1} 2^{m_\sigma^{(j'\leftarrow j)}}$, $a^{(i)} = \text{poly}(r)^{-1} 2^{m_a^{(i)}}$, $b^{(i)} = \text{poly}(r)^{-1} 2^{m_b^{(i)}}$, $\gamma = 2^{-m_\gamma} \text{poly}(r) \max_i \sigma_{2r-1\leftarrow 2i}$ and using error probability $\delta_{\mathcal{M}} \triangleq \frac{\delta}{2^{\sum_{m\in\mathcal{M}} m+1}}$. First, we note that by union bound, using the fact that $\sum_{\text{choices of } \mathcal{M}} \frac{\delta}{2^{\sum_{m\in\mathcal{M}} m+1}} = \delta$ where $\mathcal{M}$ ranges over nonnegative integers, we get that the generalization bound of Theorem C.1 holds for choices of $\mathcal{M}$ with probability $1 - \delta$.

Now for the network $F$ at hand, there would have been some choice of $\mathcal{M}$ for which the bound was applied using parameters $\hat{t}^{(i)}, \hat{\sigma}_{j'\leftarrow j}, \hat{a}^{(i)}, \hat{b}^{(i)}, \hat{\gamma}$ and

$$\|W^{(i)\top} - A^{(i)\top}\|_{2,1} \leq \hat{a}^{(i)} = \text{poly}(r)^{-1} 2^{m_a^{(i)}} \leq \text{poly}(r)^{-1} + 2\|W^{(i)\top} - A^{(i)\top}\|_{2,1}$$

$$\|W^{(i)} - B^{(i)}\|_{1,1} \leq \hat{b}^{(i)} = \text{poly}(r)^{-1} 2^{m_b^{(i)}} \leq \text{poly}(r)^{-1} + 2\|W^{(i)} - B^{(i)}\|_{1,1}$$

$$\max_{x\in P_n} \|F_{2i\leftarrow 1}(x)\| \leq \hat{t}^{(i)} = \text{poly}(r)^{-1} 2^{m_t^{(i)}} \leq \text{poly}(r)^{-1} + 2\max_{x\in P_n} \|F_{2i\leftarrow 1}\|$$

$$\max_{x\in P_n} \|Q_{j'\leftarrow j}(x)\|_{\text{op}} \leq \hat{\sigma}_{j'\leftarrow j} = \text{poly}(r)^{-1} 2^{m_\sigma^{(j'\leftarrow j)}} \leq \text{poly}(r)^{-1} + 2\max_{x\in P_n} \|Q_{j'\leftarrow j}(x)\|_{\text{op}}$$

Furthermore, using $\gamma$ to denote the true margin of the network, we also have $\hat{\gamma} \leq \gamma$ and $\frac{\hat{\sigma}_{2r-1\leftarrow 2i}}{\hat{\gamma}} \leq 4\frac{\max_{x\in P_n} \|Q_{2r-1\leftarrow 2i}(x)\|_{\text{op}}}{\gamma} + \frac{1}{\text{poly}(r)}$. Furthermore, note that the cost we pay in $\sqrt{\frac{\log(1/\delta_{\mathcal{M}})}{n}}$ is $\tilde{O}\left(r\sqrt{\frac{\log(1/\delta)}{n}}\right)$, where $\tilde{O}$ hides polylog factors in $r$ and other parameters. Thus, the bound of Theorem 5.1 holds.

The proof of the simpler Theorem 1.1, follows the same above argument. The only difference is that we union bound over parameters $\sigma, t$ and the matrix norms. $\square$

**Proposition C.1** (Dudley's entropy theorem [Dudley, 1967]). *Let $s = \max_{x \in P_n} \|x\|$ be an upper bound on the largest norm of a datapoint. Then the following bound relates Rademacher complexity to covering numbers:*

$$\mathrm{Rad}_n(\mathcal{L}) \leq \inf_{\alpha > 0} \left( \alpha + \int_\alpha^\infty \sqrt{\frac{\log \mathcal{N}(\epsilon, \mathcal{L}, s)}{n}} d\epsilon \right)$$

**Lemma C.3.** *For reference matrix $A \in \mathbb{R}^{d_1 \times d_2}$, define the class of matrices mapping functions $\mathcal{U} \triangleq \{h \mapsto Uh : U \in \mathbb{R}^{d_1 \times d_2}, \|U^\top - A^\top\|_{2,1} \leq a\}$. Then*

$$\log \mathcal{N}(\epsilon, \mathcal{U}, b) \leq \frac{2a^2 b^2}{\epsilon^2} \log(2d_1 d_2)$$

*Proof.* By Lemma 3.2 of Bartlett et al. [2017], we can construct cover $\widehat{\mathcal{U}}$ for the class $\{h \mapsto (U - A)h : U \in \mathbb{R}^{d_1 \times d_2}, \|U^\top - A^\top\|_{2,1} \leq a\}$ with the given cover size (Note that in our definition of empirical covering number, the resolution $\epsilon$ is scaled by factor $\frac{1}{n}$ versus theirs). To cover $\mathcal{U}$ with the same cardinality set, we simply shift all functions in $\widehat{\mathcal{U}}$ by $A$. $\square$

# D   Missing Proofs in Section A

We first state the proof of Theorem A.3.

*Proof of Theorem A.3.* We prove the theorem by induction on the number of non-input vertices in the vertex set $\mathcal{V}$. The statement is true if $O$ is the only non-input node in the graph: to cover the graph output with error $\epsilon_O$, we simply cover $\mathfrak{R}_O$.

Given a family of graphs $\mathcal{G}$ (with shared edges $\mathcal{E}$ and nodes $\mathcal{V}$), we assume the inductive hypothesis that "for any family of graphs with more than $|\mathcal{I}|$ input vertices, the theorem statement holds." Under this hypothesis, we will show that the theorem statement holds for the graph family $\mathcal{G}$.

We take node $V_1$ from the forest ordering $(V_1, \ldots, V_m)$ assumed in the theorem. Suppose $V_1$ depends on $C_1, \ldots, C_t$, which are assumed to be the input nodes by the definition of forest ordering. We release the node $V_1$ from the graph and obtain a new family $\mathcal{G}^{\backslash V_1} = \{G^{\backslash V_1} : G \in \mathcal{G}\}$ with a smaller number of edges than that of $\mathcal{G}$.

Define $u(h, x) \triangleq O_{G^{\backslash V_1}}(h, x)$ for $h \in \mathcal{D}_{V_1}$ and $x \in \mathcal{D}_\mathcal{I}$, and $w(x) = V_1(x)$. Then we can check that $u(w(x), x) = O_G(x)$. Let $\mathcal{U} = \{O_{G^{\backslash V_1}} : G \in \mathcal{G}\}$, and let $\mathcal{W} = \mathfrak{R}_{V_1}$. As each function in $\mathcal{U}$ is $\kappa_{V_1}$-Lipschitz in $V_1$ because of condition 1, and it equals the fixed constant $c$ if $\|V_1\| \geq s_V$ or $\|C_i\| \geq s_{C_i}$, we have $\mathcal{U}, \mathcal{W}$ satisfies the conditions of the composition lemma (see Lemma D.1). With the lemma, we conclude:

$$\log \mathcal{N}(\kappa_{V_1} \epsilon_{V_1} + \epsilon_u, \mathcal{G}, s_\mathcal{I}) \leq \log \mathcal{N}(\epsilon_u, \mathcal{U}, (s_{V_1}, s_\mathcal{I})) + \log \mathcal{N}(\epsilon_{V_1}, \mathfrak{R}_{V_1}, s_{\mathrm{pr}(V_1)}) \qquad (16)$$

Note that by the definition of forest ordering, we have that $(V_2, \ldots, V_m)$ is a forest ordering of $G^{\backslash V_1}$ and by the assumption 1 of the theorem, we have that $(V_2, \ldots, V_m)$ satisfies the condition 1 for the graph family $\mathcal{G}^{\backslash V_1}$. $\mathcal{G}^{\backslash V_1}$ has one more input node than $\mathcal{G}$, so we can invoke the inductive hypothesis on $\mathcal{G}^{\backslash V_1}$ and obtain

$$\log \mathcal{N}\left( \sum_{V \in \mathcal{V} \backslash (\{V_1, O\} \cup \mathcal{I})} \kappa_V \cdot \epsilon_V + \epsilon_O, \mathcal{U}, (s_{V_1}, s_\mathcal{I}) \right) \leq \sum_{V \in \mathcal{V} \backslash (\{V_1\} \cup \mathcal{I})} \log \mathcal{N}(\epsilon_V, \mathfrak{R}_V, s_{\mathrm{pr}(V)}) \qquad (17)$$

Combining equation (16) and (17) above, we prove (7) for $\mathcal{G}$, and complete the induction. $\square$

Below we provide the composition lemma necessary for Theorem A.3.

**Lemma D.1.** *Suppose*

$$\mathcal{U} \subseteq \{(h, x^{(1)}, \ldots, x^{(m)}) \in \mathcal{D}_h \otimes \mathcal{D}_x^{(1)} \otimes \cdots \otimes \mathcal{D}_x^{(m)} \mapsto \mathcal{D}_u\}$$

is a family of functions with two arguments and $\mathcal{W} \subseteq \{x^{(1)}, \ldots, x^{(m)} \in \mathcal{D}_x^{(1)} \otimes \cdots \otimes \mathcal{D}_x^{(m)} \mapsto \mathcal{D}_h\}$ is another family of functions. We overload notation and refer to $x^{(1)}, \ldots, x^{(m)}$ as $x$. The spaces $\mathcal{D}_h, \mathcal{D}_x, \mathcal{D}_u$ all associate with some norms $\|\cdot\|$ (the norms can potentially be different for each space, but we use the same notation for all of them.) Assume the following:

1. All functions in $\mathcal{U}$ are $\kappa$-Lipschitz in the argument $h$ for any possible choice of $x$: for any $u \in \mathcal{U}$, $x \in \mathcal{D}_x$, and $h, h' \in \mathcal{D}_h$, we have $\|u(h, x) - u(h', x)\| \leq \kappa \|h - h'\|$.

2. Any function $u \in \mathcal{U}$ collapses on inputs with large norms: there exists a constant $b$ such that $u(h, x) = b$ if $\|h\| \geq s_h$ or $\|x^{(i)}\| \geq s_x^{(i)}$ for any $i$.

Then, the family of the composition of $u$ and $w$, $\mathcal{Z} = \{z(x) = u(w(x), x) : u \in \mathcal{U}, w \in \mathcal{W}\}$, has covering number bound:

$$\log \mathcal{N}(\kappa \epsilon_w + \epsilon_u, \mathcal{Z}, s_x) \leq \log \mathcal{N}(\epsilon_w, \mathcal{W}, s_x) + \log \mathcal{N}(\epsilon_u, \mathcal{U}, (s_h, s_x))$$

*Proof.* When it is clear from context, we let $\|x\| \leq s_x$ denote the statement that $\|x^{(i)}\| \leq s_x^{(i)} \; \forall i$. Suppose $P_n$ is a uniform distribution over $n$ data points $\{x_1, \ldots, x_n\} \subset \mathcal{D}_x$ with norms not larger than $s_x$. Given function $u \in \mathcal{U}$ and $w \in \mathcal{W}$, we will construct a pair of functions such that $\hat{u}(\hat{w}(x), x)$ covers $u(w(x), x)$. We will count (in a straightforward way) how many distinct pairs of functions we have construct for all the $(u, w)$ pairs at the end of the proof.

Let $P'$ be the uniform distribution over $\{x_i : \|x_i\| \leq s_x\}$, and suppose $\hat{\mathcal{W}}$ is a $\epsilon_w \sqrt{\frac{n}{|\mathrm{supp}(P')|}}$ error cover of $\mathcal{W}$ with respect to the metric $L_2(P', \|\cdot\|)$. We note that $\hat{\mathcal{W}}$ has size at most $\mathcal{N}(\epsilon_w, \mathcal{W}, s_x)$. We found $\hat{w} \in \mathcal{W}$ such that $\hat{w}$ is $\epsilon_w$-close to $w$ in metric $L_2(P', \|\cdot\|)$. Let $\hat{h}_i$ denote $\hat{w}(x_i)$. Let $Q'$ be the uniform distribution over $\{(\hat{h}_i, x_i) : \|\hat{h}_i\| \leq s_h, \|x_i\| \leq s_x\}$, and let $Q$ be the uniform distribution over all $n$ points, $\{(\hat{h}_1, x_1), \ldots, (\hat{h}_n, x_n)\}$. Now we construct a intermediate cover $\hat{\mathcal{U}}'$ (that depends on $\hat{w}$ implicitly) that covers $\mathcal{U}$ with $\epsilon_u \sqrt{\frac{n}{|\mathrm{supp}(Q')|}}$ error with respect to the metric $L_2(Q', \|\cdot\|)$. We augment this to a cover $\hat{\mathcal{U}}$ that covers $\mathcal{U}$ with respect to metric $L_2(Q, \|\cdot\|)$ as follows: for every $\hat{u}' \in \hat{\mathcal{U}}'$, add the function $\hat{u}$ to $\hat{\mathcal{U}}$ with

$$\hat{u}(h, x) = \begin{cases} \hat{u}'(h, x) & \text{if } \|h\| \leq s_h, \|x\| \leq s_x \\ b & \text{otherwise} \end{cases}$$

Note that by construction, the size of $\hat{\mathcal{U}}$ is at most $\mathcal{N}(\epsilon_u, \mathcal{U}, (s_h, s_x))$. Now let $\hat{u}' \in \hat{\mathcal{U}}'$ be the cover element for $u$ w.r.t. $L_2(Q, \|\cdot\|)$, and $\hat{u}$ be the corresponding cover element in $\hat{\mathcal{U}}$. Because $\hat{u}(\hat{h}, x) = b = u(\hat{h}, x)$ when $\|\hat{h}\| \geq s_h$ or $\|x^{(i)}\| \geq s_x^{(i)}$ for some $i$,

$$\mathop{\mathbb{E}}_{\hat{h}, x \sim Q} \left[ \|\hat{u}(\hat{h}, x) - u(\hat{h}, x)\|^2 \right] = \frac{|\mathrm{supp}(Q')|}{n} \mathop{\mathbb{E}}_{\hat{h}, x \sim Q'} \left[ \|\hat{u}'(\hat{h}, x) - u(\hat{h}, x)\|^2 \right] \leq \epsilon_u^2 \qquad (18)$$

Then we bound the difference between $u(\hat{h}, x)$ and $u(h, x)$ by Lipschitzness; since $u(\hat{h}, x) = u(h, x) = b$ when $\|x\| > s_x$,

$$\mathop{\mathbb{E}}_{\hat{h}, x \sim Q} \left[ \|u(\hat{h}, x) - u(h, x)\|^2 \right] \leq \kappa^2 \frac{|\mathrm{supp}(P')|}{n} \mathop{\mathbb{E}}_{\hat{h}, x \sim P'} \left[ \|\hat{h} - h\|^2 \right] \leq \kappa^2 \epsilon_w^2 \qquad (19)$$

where in the last step we used the property of the cover $\mathcal{G}$. Finally, by triangle inequality, we get that

$$\|\hat{u}(\hat{w}(x), x) - u(w(x), x)\|_{L_2(P_n, \|\cdot\|)}$$
$$\leq \|\hat{u}(\hat{w}(x), x) - u(\hat{w}(x), x)\|_{L_2(P_n, \|\cdot\|)} + \|u(\hat{w}(x), x) - u(w(x), x)\|_{L_2(P_n, \|\cdot\|)}$$
$$\leq \kappa \epsilon_w + \epsilon_u \qquad \text{(by equation (18) and (19) and definition of } h_i, \hat{h}_i)$$

Finally we count how many $(\hat{w}, \hat{u})$ we have constructed: $\hat{\mathcal{W}}$ is of size at most $\mathcal{N}(\epsilon_w, \mathcal{W}, s_x)$. and for every $\hat{w} \in \hat{\mathcal{W}}$, we've constructed a family of functions $\hat{\mathcal{U}}$ (that depends on $\hat{w}$) of size at most $\mathcal{N}(\epsilon_u, \mathcal{U}, (s_h, s_x))$. Therefore, the total size of the cover is at most $\mathcal{N}(\epsilon_w, \mathcal{W}, s_x) \cdot \mathcal{N}(\epsilon_u, \mathcal{U}, (s_h, s_x))$. $\square$

## E   Missing Proofs in Section B

We first state the proofs of Theorem B.2 and Theorem B.3, which follow straightforwardly from the technical tools developed in Section F.

*Proof of Theorem B.2.* Fix any forest ordering $\mathcal{S}$ of $\widetilde{\mathcal{G}}$. Fix $\widetilde{G} \in \widetilde{\mathcal{G}}$. Let $\mathcal{S}'$ be the prefix sequence of $\mathcal{S}$ ending in $V_i$. Note that $\mathcal{S}'$ will not contain any $J_j$ or $V_j$ for $j > i$, as $V_j$ and $J_j$ will still depend on a non-input node (namely, $V_{j-1}$). Thus, we can fit $\widetilde{G}^{\backslash \mathcal{S}'}$ under the framework of Lemma F.1, where we set $k = q - i$ and identify $f_j$ with $R_{V_{i+j}}$. We set $m = i$, and identify $A_{m'}$ with $J_i \cdots J_{m'}$ (where $J_j$ may depend on input variables or itself be an input variable for $1 \leq j \leq i$, but this does not matter for our purposes). Then that to apply Lemma F.1, we set $\tau_{j' \leftarrow i'} = \kappa_{j'+i \leftarrow i'+i}$, $\tau_{j' \leftarrow 1, m'} = \kappa_{j'+i \leftarrow m'}$, and $\bar{\tau}_j = \bar{\kappa}_{i+j}$. Now we can apply Lemma F.1 to conclude that $\widetilde{G}^{\backslash \mathcal{S}'}$ is $\tilde{\kappa}_{V_i}$-Lipschitz in $V_i$ for any $1 \leq i \leq q$.

Now we prove release-Lipschitzness for a prefix sequence $\mathcal{S}'$ of $\mathcal{S}$ that ends in node $J_i$. For all $j \neq i$, fix $D_j \in \mathcal{D}_{J_j}$. It suffices to show that the function $Q$ defined by

$$Q(J_i) \triangleq \prod_{j \leq i \leq j'} \mathbb{1}_{\leq \kappa_{j' \leftarrow j}}(\|D_{j'} \cdots D_{i+1} J_i D_{i-1} \cdots D_j\|_{\mathrm{op}})$$

$$\times \prod_{j' \geq i+1} \mathbb{1}_{\leq \kappa_{j' \leftarrow i+1}}(\|D_{j'} \cdots D_{i+1}\|_{\mathrm{op}}) \times \prod_{j \leq i-1} \mathbb{1}_{\leq \kappa_{i-1 \leftarrow j}}(\|D_{i-1} \cdots D_j\|_{\mathrm{op}})$$

is $\tilde{\kappa}_{J_i}$-Lipschitz in the value of $J_i$. This is because after fixing all other inputs besides $J_i$, we can write $O_{\widetilde{G}^{\backslash \mathcal{S}'}}$ in the form $Q(J_i)a + 1$, where $a$ may depend on the other inputs but not $J_i$ and $|a| \leq 1$. Now we simply apply Lemma F.8 to conclude that $Q(J_i)$, and therefore $O_{\widetilde{G}^{\backslash \mathcal{S}'}}$, is $\tilde{\kappa}_{J_i}$-Lipschitz. □

*Proof of Theorem B.3.* We first construct an augmented family of graphs $\mathcal{G}'$ sharing the same vertices and edges as $\mathcal{G}$. For $G \in \mathcal{G}$, we add $G'$ to $\mathcal{G}'$ computing

$$O_{G'}(x) = (O_G(x) - 1) \prod_{i=1}^{q} \mathbb{1}_{\leq s_{V_i}}(\|V_i(x)\|) + 1$$

This is achieved by modifying the family of output rules as follows:

$$R'_O(x, v_1, \dots, v_q) = (R_O(x, v_1, \dots, v_q) - 1) \prod_{i=1}^{q} \mathbb{1}_{\leq s_{V_i}}(\|v_i\|) + 1$$

where $x \in \mathcal{D}_{\mathcal{I}}$ and $v_i \in \mathcal{D}_{V_i}$. We can also apply Claim J.1 to conclude that $R'_O$ outputs values in $[0, 1]$. Furthermore, as the function $\mathbb{1}_{\leq s_{V_i}}(\|v_i\|)$ is $s_{V_i}^{-1}$-Lipschitz in $v_i$, by the product property for Lipschitzness, $R'_O$ is $(c_i + s_{V_i})^{-1}$-Lipschitz in $v_i$. Now we apply Theorem B.2 to obtain a graph family $\widetilde{\mathcal{G}}$ that is $\{\tilde{\kappa}_V\}$-release-Lipschitz with respect to any forest ordering on $(\widetilde{\mathcal{V}}, \widetilde{\mathcal{E}})$ for parameters $\{\tilde{\kappa}_V\}$ defined in the theorem statement. Furthermore, by the construction of our augmentation and application of Claim J.1, it follows that

$$\tilde{R}_O(x, v_1, \dots, v_q, D_1, \dots, D_q) =$$

$$(R_O(x, v_1, \dots, v_q) - 1) \prod_{i=1}^{q} \mathbb{1}_{\leq s_{V_i}}(\|v_i\|) \prod_{1 \leq i \leq j \leq q} \mathbb{1}_{\leq \kappa_{j \leftarrow i}}(\|D_j \cdots D_i\|_{\mathrm{op}}) + 1$$

and in particular outputs the constant value 1 when $\|v_i\| > 2s_{V_i}$ or $\|D_i\| > 2\kappa_{i \leftarrow i}$. As this is a property of the output rule $\tilde{R}_O$ itself, it is clear that condition 2 of Theorem B.2 holds for any forest ordering on $(\widetilde{\mathcal{V}}, \widetilde{\mathcal{E}})$. Now we can apply Theorem B.2:

$$\log \mathcal{N}(\sum_{i \geq 1} (\tilde{\kappa}_{V_i} + \tilde{\kappa}_{J_i})\epsilon_V + \epsilon_O, O_{\widetilde{\mathcal{G}}}, s_{\mathcal{I}}) \leq \sum_{i \geq 1} \log \mathcal{N}(\epsilon_{V_i}, \mathfrak{R}_{V_i}, 2s_{V_{i-1}})$$

$$+ \log \mathcal{N}(\epsilon_{J_i}, D\mathfrak{R}_{V_i}, 2s_{V_{i-1}}) + \log \mathcal{N}(\epsilon_O, \widetilde{\mathfrak{R}}_O, \{2s_{V_i}\} \cup \{I\} \cup \{2s_{J_i}\}_{i \geq 1})$$

Now all terms match (9) except for the term $\log \mathcal{N}(\epsilon_O, \widetilde{\mathfrak{R}}_O, \{2s_{V_i}\} \cup \{I\} \cup \{2s_{J_i}\}_{i \geq 1})$. First, we note that all functions in $\widetilde{\mathfrak{R}}_O$ can be written in the form

$$\tilde{R}_O(x, v_1, \ldots, v_q, D_1, \ldots, D_q) = (R_O(x, v_1, \ldots, v_q) - 1)Q(v_1, \ldots, v_q, D_1, \ldots, D_q) + 1$$

where the function $Q$ is the same for all $\tilde{R}_O \in \widetilde{\mathfrak{R}}_O$. It follows that to cover $\widetilde{\mathfrak{R}}_O$, we can first obtain a cover $\hat{\mathfrak{R}}_O$ of $\mathfrak{R}_O$ and then apply the operation $\hat{r} \mapsto (\hat{r} - 1)Q + 1$ to each element in $\hat{\mathfrak{R}}_O$. Thus, we get the equivalence

$$\log \mathcal{N}(\epsilon_O, \widetilde{\mathfrak{R}}_O, \{2s_{V_i}\}_{i \geq 0} \cup \{2s_{J_i}\}_{i \geq 1}) = \log \mathcal{N}(\epsilon_O, \mathfrak{R}_O, \{2s_{V_i}\} \cup \{I\})$$

This allows us to conclude (9). Finally, we note that as the augmentation operations are in the form of those considered in Claim J.1, it follows that $O_{\widetilde{G}}$ upper bounds $O_G$. $\qquad\square$

## F   Technical Tools for Lipschitz Augmentation

In this section, we develop the technical tools needed for proving Theorem B.2. The main result in this section is our Lemma F.1, which essentially states that augmenting the loss with a product of Jacobians (plus additional matrices meant to model previous Jacobian nodes already released from the computational graph) will make the loss Lipschitz.

For this section, we say a function $J$ taking input $x \in \mathcal{D}$ and outputting an operator mapping $\mathcal{D}$ to $\mathcal{D}'$ is $\kappa$-Lipschitz if $\|J(x) - J(x')\|_{\mathrm{op}} \leq \kappa \|x - x'\|$ for any $x, x'$ in its input domain. We will consider functions $f_1, \ldots, f_k$, where $f_i : \mathcal{D}_{i-1} \to \mathcal{D}_i$ and $\mathcal{D}_0$ is a compact subset of some normed space. For ease of notation, we use $\| \cdot \|$ to denote the (possibly distinct) norms on $\mathcal{D}_0, \ldots, \mathcal{D}_k$. For $1 \leq i \leq j \leq k$, Let $f_{j \leftarrow i} : \mathcal{D}_{i-1} \to \mathcal{D}_j$ denote the composition

$$f_{j \leftarrow i} \triangleq f_j \circ \cdots \circ f_i$$

For convenience in indexing, for $(i, j)$ with $i > j$, we will set $f_{j \leftarrow i} : \mathcal{D}_{i-1} \to \mathcal{D}_{i-1}$ to be the identity function.

Finally consider a real-valued function $g : \mathcal{D}_0 \otimes \cdots \otimes \mathcal{D}_k \to [0, 1]$ and define the composition $z : \mathcal{D}_0 \mapsto [0, 1]$ by

$$z(x) = g(x, f_{1 \leftarrow 1}(x), \ldots, f_{k \leftarrow 1}(x))$$

We will construct a "Lipschitz-fication" for the function $z$.

Let $A_1, \ldots, A_m$ denote a collection of linear operators that map to the space $\mathcal{D}_0$. We will furthermore use $J_{j \leftarrow i, m'}$ to denote the $i$-to-$j$ Jacobian, i.e.

$$J_{j \leftarrow i, m'} \triangleq Df_{j \leftarrow i} \circ f_{i-1 \leftarrow 1}$$

When $i = 1$ and $0 \leq j \leq k$, we will also consider products between 1-to-$j$ Jacobians and the matrices $A_{m'}$: define

$$J_{j \leftarrow 1, m'} \triangleq (Df_{j \leftarrow 1})A_{m'}$$

Note in particular that $J_{0 \leftarrow 1, m'} = A_{m'}$.

**Lemma F.1.** *[Lipschitz-fication] Following the notation in this section, suppose that $g$ is $c_{k'}$-Lipschitz in its $(k' + 1)$-th argument for $0 \leq k' \leq k$. Suppose that $Df_{j \leftarrow j}$ is $\bar{\tau}_j$-Lipschitz for all $1 \leq j \leq k$. For any $(i, j)$ with $1 \leq i \leq j \leq k$, let $\tau_{j \leftarrow i}$ be parameters that intend to be a tight bound on $\|J_{j \leftarrow i}\|_{\mathrm{op}}$, and also define $\tau_{j \leftarrow 1, m'}$ which will bound $\|J_{j \leftarrow 1, m'}\|_{\mathrm{op}}$. Define the augmented function $\bar{z} : \mathcal{D}_0 \mapsto [0, 1]$ by*

$$\tilde{z}(x) = (z(x) - 1) \prod_{2 \leq i \leq j} \mathbb{1}_{\leq \tau_{j \leftarrow i}}(\|J_{j \leftarrow i}(x)\|_{\mathrm{op}}) \prod_{0 \leq j \leq k, m'} \mathbb{1}_{\leq \tau_{j \leftarrow 1, m'}}(\|J_{j \leftarrow 1, m'}(x)\|_{\mathrm{op}}) + 1$$

*Define $\tau^\star$, a Lipschitz parameter for $\tilde{z}$, by*

$$\tau^\star \triangleq \sum_{0 \leq j \leq k} 3c_j \tau_{j \leftarrow 1}$$

$$+ 18 \sum_{1 \leq i \leq j \leq k} \frac{\sum_{i'=i}^{j} \bar{\tau}_{i'} \tau_{j \leftarrow i'+1} \tau_{i'-1 \leftarrow 1} \tau_{i'-1 \leftarrow i}}{\tau_{j \leftarrow i}}$$

$$+ 18 \sum_{1 \leq j \leq k, m'} \frac{\sum_{i'=1}^{j} \bar{\tau}_{i'} \tau_{j \leftarrow i'+1} \tau_{i'-1 \leftarrow 1} \tau_{i'-1 \leftarrow 1, m'}}{\tau_{j \leftarrow 1, m'}}$$

*where for convenience we let $\tau_{j \leftarrow i} = 1$ when $j < i$. Then $\tilde{z}$ is $\tau^\star$-Lipschitz in $x$.*

*Proof.* For ease of notation, we will first define for any $(i,j)$ with $1 \leq i \leq j \leq k$, $Q_{j \leftarrow i} \triangleq \mathbb{1}_{\leq \tau_{j \leftarrow i}}(\|J_{j \leftarrow i}\|_{\text{op}})$ and for $(j, m')$ with $0 \leq j \leq k$, $Q_{j \leftarrow 1, m'} \triangleq \mathbb{1}_{\leq \tau_{j \leftarrow 1, m'}}(\|J_{j \leftarrow 1, m'}\|_{\text{op}})$. Note in particular that $Q_{0 \leftarrow 1, m'}$ is always a constant function. We will also let $\mathcal{Q}$ denote the collection of functions

$$\mathcal{Q} = \{Q_{i \leftarrow j}\}_{1 \leq i \leq j \leq k} \cup \{Q_{j \leftarrow 1, m'}\}_{0 \leq j \leq k, 1 \leq m' \leq m}$$

We define the following order $\succ_{\mathcal{Q}}$ on this collection of functions:

$$
\begin{aligned}
Q_{0 \leftarrow 1, m} &\succ_{\mathcal{Q}} \cdots \succ_{\mathcal{Q}} Q_{0 \leftarrow 1, 1} \\
&\succ_{\mathcal{Q}} Q_{1 \leftarrow 1} \succ_{\mathcal{Q}} Q_{1 \leftarrow 1, m} \succ_{\mathcal{Q}} \cdots \succ_{\mathcal{Q}} Q_{1 \leftarrow 1, 1} \\
\succ_{\mathcal{Q}} Q_{2 \leftarrow 2} &\succ_{\mathcal{Q}} Q_{2 \leftarrow 1} \succ_{\mathcal{Q}} Q_{2 \leftarrow 1, m} \succ_{\mathcal{Q}} \cdots \succ_{\mathcal{Q}} Q_{2 \leftarrow 1, 1} \\
&\qquad\qquad\qquad\qquad\qquad\qquad\qquad\qquad\vdots \\
\succ_{\mathcal{Q}} Q_{k \leftarrow k} &\succ_{\mathcal{Q}} \cdots \succ_{\mathcal{Q}} Q_{k \leftarrow 1} \succ_{\mathcal{Q}} Q_{k \leftarrow 1, m'} \succ_{\mathcal{Q}} \cdots \succ_{\mathcal{Q}} Q_{k \leftarrow 1, 1}
\end{aligned}
$$

We will first show that $\exists C > 0$ such that $\forall x$ and $\forall \nu$ with $\|\nu\| < C$, $|\tilde{z}(x) - \tilde{z}(x + \nu)| \leq \tau^{\star}\|\nu\|$. To use this statement to conclude that $\tilde{z}$ is $\tau^{\star}$-Lipschitz, we note that for arbitrary $x$ and $\nu$, we can divide the segment between $x$ and $x + \nu$ into segments of length at most $C$, and apply the above statement on each segment. First, define for $0 \leq j \leq k$

$$
\begin{aligned}
\gamma_j(x, \nu) \triangleq &g(f_{0 \leftarrow 1}(x), \ldots, f_{j-1 \leftarrow 1}(x), f_{j \leftarrow 1}(x), f_{j+1 \leftarrow 1}(x + \nu), \ldots, f_{k \leftarrow 1}(x + \nu)) \\
&- g(f_{0 \leftarrow 1}(x), \ldots, f_{j-1 \leftarrow 1}(x), f_{j \leftarrow 1}(x + \nu), f_{j+1 \leftarrow 1}(x + \nu), \ldots, f_{k \leftarrow 1}(x + \nu))
\end{aligned}
$$

Next, define the telescoping differences

$$\delta_j(x, \nu) \triangleq \gamma_j(x, \nu) \prod_{Q \succeq_{\mathcal{Q}} Q_{j \leftarrow 1, 1}} Q(x) \prod_{Q_{j \leftarrow 1, 1} \succ_{\mathcal{Q}} Q} Q(x + \nu) \; \forall 0 \leq j \leq k \tag{20}$$

$$\Delta_{j \leftarrow i}(x, \nu) \triangleq (Q_{j \leftarrow i}(x) - Q_{j \leftarrow i}(x + \nu)) \prod_{Q \succ_{\mathcal{Q}} Q_{j \leftarrow i}} Q(x) \prod_{Q_{j \leftarrow i} \succ_{\mathcal{Q}} Q} Q(x + \nu) \; \forall 1 \leq i \leq j \leq k \tag{21}$$

$$
\begin{aligned}
\Delta_{j \leftarrow 1, m'}(x, \nu) \triangleq &(Q_{j \leftarrow 1, m'}(x) - Q_{j \leftarrow 1, m'}(x + \nu)) \cdot \\
&\prod_{Q \succ_{\mathcal{Q}} Q_{j \leftarrow 1, m'}} Q(x) \prod_{Q_{j \leftarrow 1, m'} \succ_{\mathcal{Q}} Q} Q(x + \nu) \; \forall 0 \leq j \leq k
\end{aligned}
\tag{22}
$$

Now note that by Claim F.7, we have the bound

$$|\tilde{z}(x) - \tilde{z}(x + \nu)| \leq \sum_{0 \leq j \leq k} |\delta_j(x, \nu)| + \sum_{1 \leq i \leq j \leq k} |\Delta_{j \leftarrow i}(x, \nu)| + \sum_{0 \leq j \leq k, m'} |\Delta_{j \leftarrow 1, m'}(x, \nu)|$$

Define $\bar{\tau}$ to be the Lipschitz constant of $J_{j \leftarrow i}$ on $\mathcal{D}_0$ for all $1 \leq i \leq j \leq k$ guaranteed by Claim F.6. First, note that $\Delta_{0 \leftarrow 1, m'} = 0$ for all $m'$. Thus, by Claims F.4 and F.5, it follows that

$$
\begin{aligned}
|\tilde{z}(x) - \tilde{z}(x + \nu)| \leq &\sum_{0 \leq j \leq k} c_j (2\tau_{j \to 1} + \frac{\bar{\tau}}{2}\|\nu\|)\|\nu\| \\
&+ \sum_{1 \leq i \leq j \leq k} \|\nu\| \frac{\sum_{i'=i}^{j} (2\tau_{j \leftarrow i'+1} + \bar{\tau}\|\nu\|)\bar{\tau}_{i'}(2\tau_{i'-1 \leftarrow 1} + \frac{\bar{\tau}}{2}\|\nu\|)2\tau_{i'-1 \leftarrow i}}{\tau_{j \leftarrow i}} \\
&+ \sum_{1 \leq j \leq k, 1 \leq m' \leq m} \|\nu\| \frac{\sum_{i'=1}^{j} (2\tau_{j \leftarrow i'+1} + \bar{\tau}\|\nu\|)\bar{\tau}_{i'}(2\tau_{i'-1 \leftarrow 1} + \frac{\bar{\tau}}{2}\|\nu\|)2\tau_{i'-1 \leftarrow 1, m'}}{\tau_{j \leftarrow 1, m'}}
\end{aligned}
\tag{23}
$$

Now note that if $\|\nu\| \leq \frac{2\min_{i\leq j}\tau_{j\leftarrow i}}{\bar{\tau}}$, then it follows that $2\tau_{j\leftarrow i} + \frac{\bar{\tau}}{2}\|\nu\| \leq 3\tau_{j\leftarrow i}\forall i \leq j$. Substituting into (23), we get that $\forall x, \|\nu\| \leq \frac{2\min_{i\leq j}\tau_{j\leftarrow i}}{\bar{\tau}}$,

$$
\begin{aligned}
|\tilde{z}(x) - \tilde{z}(x+\nu)| \leq & \|\nu\| \sum_{0\leq j\leq k} 3c_j\tau_{j\leftarrow 1} \\
& + \|\nu\|18 \sum_{1\leq i\leq j\leq k} \frac{\sum_{i'=i}^{j}\bar{\tau}_{i'}\tau_{j\leftarrow i'+1}\tau_{i'-1\leftarrow 1}\tau_{i'-1\leftarrow i}}{\tau_{j\leftarrow i}} \\
& + \|\nu\|18 \sum_{1\leq j\leq k, m'} \frac{\sum_{i'=1}^{j}\bar{\tau}_{i'}\tau_{j\leftarrow i'+1}\tau_{i'-1\leftarrow 1}\tau_{i'-1\leftarrow 1,m'}}{\tau_{j\leftarrow 1,m'}} \\
= & \tau^{\star}\|\nu\|
\end{aligned}
$$

It follows that $\tilde{z}$ is $\tau^{\star}$-Lipschitz. $\qquad\square$

**Claim F.2.** *In the setting of Lemma F.1, for $1 \leq i \leq j \leq k$, we can expand the error $J_{j\leftarrow i}(x) - J_{j\leftarrow i}(x+\nu)$ as follows:*

$$
J_{j\leftarrow i}(x) - J_{j\leftarrow i}(x+\nu) = \sum_{i'=i}^{j} J_{j\leftarrow i'+1}(x+\nu)(J_{i'\leftarrow i'}(x) - J_{i'\leftarrow i'}(x+\nu))J_{i'-1\leftarrow i}(x) \quad (24)
$$

*Furthermore, for $1 \leq j \leq k, m'$, we can expand the error $J_{j\leftarrow 1,m'}(x) - J_{j\leftarrow 1,m'}(x+\nu)$ as follows:*

$$
J_{j\leftarrow 1,m'}(x) - J_{j\leftarrow 1,m'}(x+\nu) = \sum_{i'=1}^{j} J_{j\leftarrow i'+1}(x+\nu)(J_{i'\leftarrow i'}(x) - J_{i'\leftarrow i'}(x+\nu))J_{i'-1\leftarrow 1,m'}(x)
$$
$$(25)$$

*Proof.* We will first show (24) by inducting on $j - i$. The base case $j = i$ follows by definition, as we can reduce $J_{i\leftarrow i+1}$ and $J_{i-1\leftarrow i}$ to constant-valued functions that output the identity matrix.

For the inductive step, we use Claim J.2 to expand

$$
\begin{aligned}
J_{j\leftarrow i}(x) - J_{j\leftarrow i}(x+\nu) = & J_{j\leftarrow i+1}(x)J_{i\leftarrow i}(x) - J_{j\leftarrow i+1}(x+\nu)J_{i\leftarrow i}(x+\nu) \\
= & (J_{j\leftarrow i+1}(x) - J_{j\leftarrow i+1}(x+\nu))J_{i\leftarrow i}(x) \\
& + J_{j\leftarrow i+1}(x+\nu)(J_{i\leftarrow i}(x) - J_{i\leftarrow i}(x+\nu)) \\
= & \sum_{i'=i+1}^{j} J_{j\leftarrow i'+1}(x+\nu)(J_{i'\leftarrow i'}(x) - J_{i'\leftarrow i'}(x+\nu))J_{i'-1\leftarrow i+1}(x)J_{i\leftarrow i}(x) \\
& \hspace{4cm} \text{(by the inductive hypothesis)} \\
& + J_{j\leftarrow i+1}(x+\nu)(J_{i\leftarrow i}(x) - J_{i\leftarrow i}(x+\nu)) \\
= & \sum_{i'=i+1}^{j} J_{j\leftarrow i'+1}(x+\nu)(J_{i'\leftarrow i'}(x) - J_{i'\leftarrow i'}(x+\nu))J_{i'-1\leftarrow i}(x) \\
& \hspace{4cm} \text{(by Claim J.2)} \\
& + J_{j\leftarrow i+1}(x+\nu)(J_{i\leftarrow i}(x) - J_{i\leftarrow i}(x+\nu)) \\
= & \sum_{i'=i}^{j} J_{j\leftarrow i'+1}(x+\nu)(J_{i'\leftarrow i'}(x) - J_{i'\leftarrow i'}(x+\nu))J_{i'-1\leftarrow i}(x)
\end{aligned}
$$

as desired.

To prove (25), we first note that by definition, $J_{j\leftarrow 1,m'}(x) = J_{j\leftarrow 1}(x)J_{0\leftarrow 1,m'}$, so

$$J_{j\leftarrow 1,m'}(x) - J_{j\leftarrow 1,m'}(x+\nu) \tag{26}$$

$$= (J_{j\leftarrow 1}(x) - J_{j\leftarrow 1}(x+\nu))J_{0\leftarrow 1,m'}$$

$$= \sum_{i'=1}^{j} J_{j\leftarrow i'+1}(x+\nu)(J_{i'\leftarrow i'}(x) - J_{i'\leftarrow i'}(x+\nu))J_{i'-1\leftarrow 1}(x)J_{0\leftarrow 1,m'} \qquad \text{(by (24))}$$

$$= \sum_{i'=1}^{j} J_{j\leftarrow i'+1}(x+\nu)(J_{i'\leftarrow i'}(x) - J_{i'\leftarrow i'}(x+\nu))J_{i'-1\leftarrow 1,m'}(x)$$

$$\text{(since } J_{i'-1\leftarrow 1}(x)J_{0\leftarrow 1,m'} = J_{i'-1\leftarrow 1,m'}(x))$$

$\square$

**Claim F.3.** *In the setting of Lemma F.1, suppose that $J_{j\leftarrow i}$ is $\bar{\tau}$-Lipschitz for all $1 \leq i \leq j \leq k$. Then we can bound the operator norm error in the Jacobian by*

$$\|J_{j\leftarrow i}(x) - J_{j\leftarrow i}(x+\nu)\|_{\mathrm{op}} \leq$$

$$\|\nu\| \sum_{i'=i}^{j} (\|J_{j\leftarrow i'+1}(x)\|_{\mathrm{op}} + \bar{\tau}\|\nu\|)\bar{\tau}_{i'}(\|J_{i'-1\leftarrow 1}(x)\|_{\mathrm{op}} + \frac{\bar{\tau}}{2}\|\nu\|)\|J_{i'-1\leftarrow i}(x)\|_{\mathrm{op}} \tag{27}$$

*Likewise, we can bound the operator norm error in the product between Jacobian and auxiliary matrices by*

$$\|J_{j\leftarrow 1,m'}(x) - J_{j\leftarrow 1,m'}(x+\nu)\|_{\mathrm{op}} \leq$$

$$\|\nu\| \sum_{i'=1}^{j} (\|J_{j\leftarrow i'+1}(x)\|_{\mathrm{op}} + \bar{\tau}\|\nu\|)\bar{\tau}_{i'}(\|J_{i'-1\leftarrow 1}(x)\|_{\mathrm{op}} + \frac{\bar{\tau}}{2}\|\nu\|)\|J_{i'-1\leftarrow 1,m'}(x)\|_{\mathrm{op}} \tag{28}$$

*Proof.* We will first prove (27), as the proof of (28) is nearly identical. Starting from (24) of Claim F.2, we have

$$J_{j\leftarrow i}(x) - J_{j\leftarrow i}(x+\nu) = \sum_{i'=i}^{j} J_{j\leftarrow i'+1}(x+\nu)(J_{i'\leftarrow i'}(x) - J_{i'\leftarrow i'}(x+\nu))J_{i'-1\leftarrow i}(x)$$

By triangle inequality and the fact that $J_{j'\leftarrow i'}$ is $\bar{\tau}$-Lipschitz $\forall i' \leq j'$, it follows that

$$\|J_{j\leftarrow i}(x) - J_{j\leftarrow i}(x+\nu)\|_{\mathrm{op}} \tag{29}$$

$$\leq \sum_{i'=i}^{j} \|J_{j\leftarrow i'+1}(x+\nu)\|_{\mathrm{op}}\|J_{i'\leftarrow i'}(x) - J_{i'\leftarrow i'}(x+\nu)\|_{\mathrm{op}}\|J_{i'-1\leftarrow i}(x)\|_{\mathrm{op}}$$

$$\leq \sum_{i'=i}^{j} (\|J_{j\leftarrow i'+1}(x)\|_{\mathrm{op}} + \bar{\tau}\|\nu\|)\|J_{i'\leftarrow i'}(x) - J_{i'\leftarrow i'}(x+\nu)\|_{\mathrm{op}}\|J_{i'-1\leftarrow i}(x)\|_{\mathrm{op}} \tag{30}$$

Next, we note that

$$\|J_{i'\leftarrow i'}(x) - J_{i'\leftarrow i'}(x+\nu)\|_{\mathrm{op}} = \|Df_{i'\leftarrow i'}[f_{i'-1\leftarrow 1}(x)] - Df_{i'\leftarrow i'}[f_{i'-1\leftarrow 1}(x+\nu)]\|_{\mathrm{op}}$$

$$\leq \bar{\tau}_{i'}\|f_{i'-1\leftarrow 1}(x) - f_{i'-1\leftarrow 1}(x+\nu)\|$$

$$\leq \bar{\tau}_{i'}(\|J_{i'-1\leftarrow 1}(x)\|_{\mathrm{op}} + \frac{\bar{\tau}}{2}\|\nu\|)\|\nu\| \qquad \text{(applying Claim J.4)}$$

Plugging the above into (30), we get (27). To prove (28), we start from (25) and follow the same steps as above. $\square$

**Claim F.4.** *In the setting of Lemma F.1, suppose that $J_{j\leftarrow i}$ is $\bar{\tau}$-Lipschitz for all $1 \leq i \leq j \leq k$. Then we can upper bound the error terms corresponding to the indicators by*

$$|\Delta_{j\leftarrow i}(x,\nu)| \leq \|\nu\| \frac{\sum_{i'=i}^{j}(2\tau_{j\leftarrow i'+1} + \bar{\tau}\|\nu\|)\bar{\tau}_{i'}(2\tau_{i'-1\leftarrow 1} + \frac{\bar{\tau}}{2}\|\nu\|)2\tau_{i'-1\leftarrow i}}{\tau_{j\leftarrow i}} \tag{31}$$

*Likewise, the following upper bound holds for all $(j,m')$ with $1 \leq j \leq k, 1 \leq m' \leq m$:*

$$|\Delta_{j\leftarrow 1,m'}(x,\nu)| \leq \|\nu\| \frac{\sum_{i'=1}^{j}(2\tau_{j\leftarrow i'+1} + \bar{\tau}\|\nu\|)\bar{\tau}_{i'}(2\tau_{i'-1\leftarrow 1} + \frac{\bar{\tau}}{2}\|\nu\|)2\tau_{i'-1\leftarrow 1,m'}}{\tau_{j\leftarrow 1,m'}} \tag{32}$$

*Proof.* We will prove (31) as the proof of (32) is analogous. Note that as $\mathbb{1}_{\leq \tau_{j \leftarrow i}}$ is $\frac{1}{\tau_{j \leftarrow i}}$-Lipschitz in its argument, we have

$$|Q_{j \leftarrow i}(x) - Q_{j \leftarrow i}(x + \nu)| = |\mathbb{1}_{\leq \tau_{j \leftarrow i}}(\|J_{j \leftarrow i}(x)\|_{\mathrm{op}}) - \mathbb{1}_{\leq \tau_{j \leftarrow i}}(\|J_{j \leftarrow i}(x + \nu)\|_{\mathrm{op}})|$$

$$\leq \frac{1}{\tau_{j \leftarrow i}} |\|J_{j \leftarrow i}(x)\|_{\mathrm{op}} - \|J_{j \leftarrow i}(x + \nu)\|_{\mathrm{op}}|$$

$$\leq \frac{1}{\tau_{j \leftarrow i}} \|J_{j \leftarrow i}(x) - J_{j \leftarrow i}(x + \nu)\|_{\mathrm{op}}$$

Plugging this into our definition for $\Delta_{j \leftarrow i}$ (21), it follows that

$$|\Delta_{j \leftarrow i}(x, \nu)| \leq \frac{1}{\tau_{j \leftarrow i}} \|J_{j \leftarrow i}(x) - J_{j \leftarrow i}(x + \nu)\|_{\mathrm{op}} \prod_{Q \succ_{\mathcal{Q}} Q_{j \leftarrow i}} Q(x) \prod_{Q_{j \leftarrow i} \succ_{\mathcal{Q}} Q} Q(x + \nu) \quad (33)$$

Now we define the set $\mathcal{E}$ by

$$\mathcal{E} = \cap_{i \leq i' \leq j} \{x : \|J_{j \leftarrow i'+1}(x)\|_{\mathrm{op}} \leq 2\tau_{j \leftarrow i'+1}, \|J_{i'-1 \leftarrow 1}(x)\|_{\mathrm{op}} \leq 2\tau_{i'-1 \leftarrow 1},$$
$$\text{and } \|J_{i'-1 \leftarrow i}(x)\|_{\mathrm{op}} \leq 2\tau_{i'-1 \leftarrow i}\}$$

Note that if $x \notin \mathcal{E}$, then $\exists i' < j'$ such that $Q_{j' \leftarrow i'}(x) = 0$ and $Q_{j' \leftarrow i'} \succ_{\mathcal{Q}} Q_{j \leftarrow i}$ by definition of the order $\succ_{\mathcal{Q}}$. It follows that if $x \notin \mathcal{E}$, $\prod_{h \succ_{\mathcal{Q}} Q_{j \leftarrow i}} h(x) = 0$, so $|\Delta_{j \leftarrow i}(x, \nu)| = 0$. Otherwise, if $x \in \mathcal{E}$, by Claim F.3 we have

$$\|J_{j \leftarrow i}(x) - J_{j \leftarrow i}(x + \nu)\|_{\mathrm{op}} \leq \|\nu\| \sum_{i'=i}^{j} (2\tau_{j \leftarrow i'+1} + \bar{\tau}\|\nu\|)\bar{\tau}_{i'}(2\tau_{i'-1 \leftarrow 1} + \frac{\bar{\tau}}{2}\|\nu\|)2\tau_{i'-1 \leftarrow i}$$

where we recall that $\tau_{i-1 \leftarrow i} = 1$. Plugging this into (33) and using the fact that all functions $h \in \mathcal{Q}$ are bounded by 1 gives the desired statement.

To prove (32), we simply apply the above argument with (28). $\qquad\square$

**Claim F.5.** *In the setting of Lemma F.1, fix index $j$ with $0 \leq j \leq k$ and suppose that $J_{j \leftarrow 1}$ is $\bar{\tau}$-Lipschitz. Then we can bound the error due to function composition by*

$$|\delta_j(x, \nu)| \leq c_j (2\tau_{j \rightarrow 1} + \frac{\bar{\tau}}{2}\|\nu\|)\|\nu\|$$

*Proof.* Starting from (20), we can first express $\delta_i(x, \nu)$ by

$$\delta_j(x, \nu) = \gamma_j(x, \nu) Q_{j \leftarrow 1}(x) \prod_{Q \succeq_{\mathcal{Q}} Q_{j \leftarrow 1,1}, Q \neq Q_{j \leftarrow 1}} Q(x) \prod_{Q_{j \leftarrow 1,1} \succ_{\mathcal{Q}} Q} Q(x + \nu)$$

as $Q_{j \leftarrow 1} \succ_{\mathcal{Q}} Q_{j \leftarrow 1,1}$. First we note that by definition, $|\gamma_j(x, \nu)| \leq c_j \|f_{j \leftarrow 1}(x) - f_{j \leftarrow 1}(x + \nu)\|$, as the function $g$ is $c_j$-Lipschitz in its $j$-th argument. Thus, since all functions $Q \in \mathcal{Q}$ are bounded by 1, it follows that

$$|\delta_j(x, \nu)| \leq |\gamma_j(x, \nu)| Q_{j \leftarrow 1}(x)$$
$$\leq c_j \|f_{j \leftarrow 1}(x) - f_{j \leftarrow 1}(x + \nu)\| \mathbb{1}_{\leq \tau_{j \leftarrow 1}}(\|J_{j \leftarrow 1}(x)\|_{\mathrm{op}})$$
$$\leq c_j (2\tau_{j \rightarrow 1} + \frac{\bar{\tau}}{2}\|\nu\|)\|\nu\| \qquad \text{(by Claim J.4)}$$

$\qquad\square$

**Claim F.6.** *In the setting of Lemma F.1, $\exists \bar{\tau}$ such that $\forall i \leq j$, $J_{j \leftarrow i}$ is $\bar{\tau}$-Lipschitz on a compact domain $\mathcal{D}_0$.*

*Proof.* We first show inductively that $f_{i \leftarrow 1}$ is Lipschitz for all $i$. The base case $f_{1 \leftarrow 1}$ follows by definition, as $f_{1 \leftarrow 1}$ is continuously differentiable and $\mathcal{D}_0$ is a compact set.

Now we show the inductive step: first write $f_{i \leftarrow 1} = f_i \circ f_{i-1 \leftarrow 1}$. By continuity, $\{f_{i-1 \leftarrow 1}(x) : x \in \mathcal{D}_0\}$ is compact. Furthermore, $f_i$ is continuously differentiable under the assumptions of Lemma F.1.

Thus, $f_i$ is Lipschitz on domain $\{f_{i-1\leftarrow 1}(x) : x \in \mathcal{D}_0\}$. As $f_{i\leftarrow 1} = f_i \circ f_{i-1\leftarrow 1}$ is the composition of Lipschitz functions by the inductive hypothesis, $f_{i\leftarrow 1}$ is itself Lipschitz.

Now it follows that $\forall i$, $J_{i\leftarrow i}$ is Lipschitz on $\mathcal{D}_0$, as it is the composition of $Df_{i\leftarrow i}$ and $f_{i-1\leftarrow 1}$, both of which are Lipschitz. Finally, by the chain rule (Claim J.2), we have that $J_{j\leftarrow i} = J_{j\leftarrow j} \cdots J_{i\leftarrow i}$ is the product of Lipschitz functions, and therefore Lipschitz for all $i < j$. We simply take $\bar{\tau}$ to be the maximum Lipschitz constant of $J_{j\leftarrow i}$ over all $i \leq j$. $\qquad\square$

**Claim F.7.** *In the setting of Lemma F.1,*

$$|\tilde{z}(x) - \tilde{z}(x+\nu)| \leq \sum_{0 \leq j \leq k} |\delta_j(x,\nu)| + \sum_{1 \leq i \leq j \leq k} |\Delta_{j\leftarrow i}(x,\nu)| + \sum_{0 \leq j \leq k, m'} |\Delta_{j-1,m'}(x,\nu)|$$

*Proof.* For $0 \leq j \leq k+1$, define $z_j(x,\nu)$ by

$$z_j(x,\nu) \triangleq g(f_{0\leftarrow 1}(x), \ldots, f_{j-1\leftarrow 1}(x), f_{j\leftarrow 1}(x+\nu), f_{j+1\leftarrow 1}(x+\nu), \ldots, f_{k\leftarrow 1}(x+\nu))$$

Thus, $z_j(x,\nu)$ denotes $g \circ (f_{0\leftarrow 1} \otimes \ldots \otimes f_{k\leftarrow 1})$ with the last $k+1-j$ inputs to $g$ depending on $x+\nu$ instead of $x$. Now we claim that by a telescoping argument (Claim J.3),

$$\tilde{z}(x) - \tilde{z}(x+\nu) =$$
$$\sum_{0 \leq j \leq k} \delta_j(x,\nu) + \sum_{1 \leq i \leq j \leq k} (z_k(j,\nu) - 1)\Delta_{j\leftarrow i} + \sum_{0 \leq j \leq k, m'} (z_j(x,\nu) - 1)\Delta_{j-1,m'} \qquad (34)$$

To see this, compute the sum in the order the following sequence of terms, which corresponds to a traversal of $\mathcal{Q}$ in least-to-greatest order:

$$\delta_k, (z_k(x,\nu) - 1)\Delta_{k-1,1}, \ldots, (z_k(x,\nu) - 1)\Delta_{k-1,m'}, (z_k(x,\nu) - 1)\Delta_{k\leftarrow 1}, \ldots, (z_k(x,\nu) - 1)\Delta_{k\leftarrow k}$$

$$\vdots$$

$$\delta_1, (z_1(x,\nu) - 1)\Delta_{1-1,1}, \ldots, (z_1(x,\nu) - 1)\Delta_{1-1,m'}, (z_1(x,\nu) - 1)\Delta_{1\leftarrow 1}$$
$$\delta_0, (z_0(x,\nu) - 1)\Delta_{0-1,1}, \ldots, (z_0(x,\nu) - 1)\Delta_{0-1,m'}$$

Now we simply apply triangle inequality on (34) and use the fact that $z_j(x,\nu) - 1 \in [-1, 0]$ $\forall 0 \leq j \leq k+1$ to obtain the desired statement. $\qquad\square$

**Lemma F.8.** *In the setting of Theorem B.2, fix $1 \leq i \leq p$ and define*

$$Q(J_i) \triangleq \prod_{j \leq i \leq j'} \mathbb{1}_{\leq \kappa_{j'\leftarrow j}}(\|D_{j'} \cdots D_{i+1} J_i D_{i-1} \cdots D_j\|_{\mathrm{op}})$$

$$\times \prod_{j' \geq i+1} \mathbb{1}_{\leq \kappa_{j'\leftarrow i+1}}(\|D_{j'} \cdots D_{i+1}\|_{\mathrm{op}}) \times \prod_{j \leq i-1} \mathbb{1}_{\leq \kappa_{i-1\leftarrow j}}(\|D_{i-1} \cdots D_j\|_{\mathrm{op}})$$

*Then $Q$ is $\tilde{\kappa}_{J_i}$-Lipschitz in $J_i$, where*

$$\tilde{\kappa}_{J_i} \triangleq \sum_{j \leq i \leq j'} \frac{4\kappa_{j'\leftarrow i+1}\kappa_{i-1\leftarrow j}}{\kappa_{j'\leftarrow j}}$$

*Here for convenience we use the convention that $\kappa_{i-1\leftarrow i} = 1$.*

*Proof.* There are two cases: the condition $\|D_{j'} \cdots D_{i+1}\|_{\mathrm{op}} \leq 2\kappa_{j'\leftarrow i+1}$ and $\|D_{i-1} \cdots D_j\|_{\mathrm{op}} \leq 2\kappa_{i-1\leftarrow j}$ for all $j' \geq i+1$, $j \leq i-1$ either holds or does not hold. In the case that it does not hold, $Q$ is the constant function at 0, and is certainly $\tilde{\kappa}_{J_i}$-Lipschitz. In the case that the condition does hold, $\mathbb{1}_{\leq \kappa_{j'\leftarrow j}}(\|D_{j'} \cdots D_{i+1} J_i D_{i-1} \cdots D_j\|_{\mathrm{op}})$ is $\frac{\kappa_{j'\leftarrow i+1}\kappa_{i-1\leftarrow j}}{\kappa_{j'\leftarrow j}}$-Lipschitz for all $j' \leq i \leq j$, and therefore their product is $\tilde{\kappa}_{J_i}$-Lipschitz. As the remaining indicators that do not depend on $J_i$ are constants in $[0, 1]$, it follows that $Q$ is $\tilde{\kappa}_{J_i}$-Lipschitz. $\qquad\square$

# G Application to Recurrent Neural Networks

In this section, we will apply our techniques to recurrent neural networks. Suppose that we are in a classification setting. For simplicity, we will assume that the hidden layer and input dimensions are $d$. We will define a recurrent neural network with $r-1$ activation layers as follows using parameters $W, U, Y$, activation $\phi$ and input sequence $x = (x^{(0)}, \ldots, x^{(r-2)})$:

$$F(x) = Y h^{(2r-2)}(x)$$
$$h^{(2i)}(x) = \phi(h^{(2i-1)}(x) + u^{(i-1)}(x))$$
$$h^{(2i-1)}(x) = W h^{(2i-2)}(x)$$
$$u^{(i-1)}(x) = U x^{(i-1)}$$

where $h^{(0)}$ is set to be 0. Now following the convention of Section 5, we will define the interlayer Jacobians. For odd indices $2i-1$, $i \leq r-1$, we simply set $Q_{2i-1\leftarrow 2i-1}$ to the constant function $x \mapsto W$. For even indices $2i$, $i \leq r-1$, we set $Q_{2i\leftarrow 2i}(x) \triangleq D\phi[h^{2i-1}(x) + u^{(i-1)}(x)]$, the Jacobian of the activation applied to the input of $h^{(2i)}(x)$. Finally, we set $Q_{2r-1\leftarrow 2r-1}$ to be the constant function $x \mapsto Y$. Now for $i' > i$, we set $Q_{i'\leftarrow i}(x) = Q_{i'\leftarrow i'}(x) \cdots Q_{i\leftarrow i}(x)$. If $i' < i$, we set $Q_{i'\leftarrow i}$ to the identity matrix.

With this notation in place, we can state our generalization bound for RNN's:

**Theorem G.1.** *Assume that the activation $\phi$ is 1-Lipschitz with a $\bar{\sigma}_\phi$-Lipschitz derivative. With probability $1-\delta$ over the random draws of $P_n$, all RNNs $F$ will satisfy the following generalization guarantee:*

$$\mathop{\mathbb{E}}_{(x,y)\sim P}[l_{0\text{-}1}(F(x),y)] \leq$$

$$\tilde{O}\left(\frac{\left((\kappa^{\text{rnn-hidden},(r)} a_Y t^{(r-1)})^{2/3} + \sum_{i=1}^{r-1} \kappa^{\text{rnn-hidden},(i)\,2/3}((a_W t^{(i-1)})^{2/3} + (a_U t^{\text{data}})^{2/3}) + \sum_{i=1}^{r}(\kappa^{\text{rnn-jacobian},(i)} b)^{2/3}\right)^{3/2}}{\sqrt{n}}\right)$$

$$+\tilde{O}\left(r\sqrt{\frac{\log(1/\delta)}{n}}\right)$$

*where $\kappa^{\text{jacobian},(i)} \triangleq \sum_{1\leq j\leq 2i-1\leq j'\leq 2r-1} \frac{\sigma_{j'\leftarrow 2i}\sigma_{2i-2\leftarrow j}}{\sigma_{j'\leftarrow j}}$, and*

$$\kappa^{\text{hidden},(i)} \triangleq \frac{1}{\text{poly}(r)} + \frac{\sigma_{2r-1\leftarrow 2i}}{\gamma} + \sum_{i\leq i'<r}\frac{\sigma_{2i'\leftarrow 2i}}{t^{(i')}} + \sum_{1\leq j\leq j'\leq 2r-1}\sum_{\substack{j''=\max\{2i,j\},\\ j''\text{ even}}}^{j'} \frac{\bar{\sigma}_\phi \sigma_{j'\leftarrow j''+1}\sigma_{j''-1\leftarrow 2i}\sigma_{j''-1\leftarrow j}}{\sigma_{j'\leftarrow j}}$$

*In these expressions, we define $\sigma_{j-1\leftarrow j} = 1$, and:*

$$a_W \triangleq \text{poly}(r)^{-1} + \|W^\top\|_{2,1}, a_U \triangleq \text{poly}(r)^{-1} + \|U^\top\|_{2,1}$$

$$a_Y \triangleq \text{poly}(r)^{-1} + \|Y^\top\|_{2,1}, b \triangleq \text{poly}(r)^{-1} + \|W\|_{1,1}$$

$$t^{(0)} = 0, t^{\text{data}} \triangleq \max_{x\in P_n}\max_i \|x^{(i)}\|, \ t^{(i)} \triangleq \text{poly}(r)^{-1} + \max_{x\in P_n} \|h^{(2i)}(x)\|$$

$$\sigma_{j'\leftarrow j} \triangleq \text{poly}(r)^{-1} + \max_{x\in P_n} \|Q_{j'\leftarrow j}(x)\|_{\text{op}}, \text{ and } \gamma \triangleq \min_{(x,y)\in P_n} [F(x)]_y - \max_{y'\neq y}[F(x)]_{y'} > 0$$

*Note that the training error here is 0 because of the existence of positive margin $\gamma$.*

Our proof follows the template of Theorem 5.1: we bound the Rademacher complexity of some augmented RNN loss. We then argue for generalization of the augmented loss and perform a union bound over all the choices of parameters. As the latter steps are identical to those in the proof of Theorem 5.1, we omit these and focus on bounding the Rademacher complexity of an augmented RNN loss.

**Theorem G.2.** *Suppose that $\phi$ is 1-Lipschitz with $\bar{\sigma}_\phi$-Lipschitz derivative. Define the following class of RNNs with bounded weight matrices:*

$$\mathcal{F} \triangleq \left\{x \mapsto F(x) : \|W^\top\|_{2,1} \leq a_Y, \|U^\top\|_{2,1} \leq a_U, \|Y^\top\|_{2,1} \leq a_Y, \|W\|_{1,1} \leq b, \|W\|_{\text{op}} \leq \sigma_W, \|Y\|_{\text{op}} \leq \sigma_Y\right\}$$

*and let $\sigma_{j'\leftarrow j}$ be parameters that will bound the $j$ to $j'$ layerwise Jacobian for $j' \geq j$, where we set $\sigma_{2i\leftarrow 2i} = 1$ and $\sigma_{2i-1\leftarrow 2i-1} = \sigma_W$ for $i \leq r-1$, $\sigma_{2r-1\leftarrow 2r-1} = \sigma_Y$. Let $t^{(i)}$ be parameters bounding the layer norm after applying the $i$-th activation, and let $t^{(0)} = 0$, $t^{\text{data}} = \max_{x\in P_n} \max_i \|x^{(i)}\|$. Define the class of augmented losses*

$$\mathcal{L}_{\text{rnn-aug}} \triangleq \left\{ (l_\gamma - 1) \circ F \prod_{i=1}^{r-1} \mathbb{1}_{\leq t^{(i)}}(\|h^{(2i)}\|) \prod_{1\leq j<j'\leq 2r-1} \mathbb{1}_{\leq \sigma_{j'\leftarrow j}}(\|Q_{j'\leftarrow j}\|_{\text{op}}) + 1 : F \in \mathcal{F} \right\}$$

*and define for $1 \leq i \leq r$, $\kappa^{\text{jacobian},(i)}$, $\kappa^{\text{hidden},(i)}$ meant to bound the influence of the matrix $W^{(i)}$ on the Jacobians and hidden variables, respectively as in* (11), (12). *Then we can bound the empirical Rademacher complexity of the augmented loss class by*

$$\text{Rad}_n(\mathcal{L}_{\text{rnn-aug}}) =$$

$$\tilde{O}\left( \frac{\left((\kappa^{\text{rnn-hidden},(r)} a_Y t^{(r-1)})^{2/3} + \sum_{i=1}^{r-1} \kappa^{\text{rnn-hidden},(i)\,2/3}((a_W t^{(i-1)})^{2/3} + (a_U t^{\text{data}})^{2/3}) + \sum_{i=1}^{r}(\kappa^{\text{rnn-jacobian},(i)} b)^{2/3}\right)^{3/2}}{\sqrt{n}} \right)$$

*where $\kappa^{\text{rnn-hidden},(i)}$, $\kappa^{\text{rnn-jacobian},(i)}$ are defined in Theorem G.1.*

*Proof.* We will associate the family of losses $\mathcal{L}_{\text{rnn-aug}}$ with a computational graph structure on internal nodes $H_1, H_2, \ldots, H_{2r-1}, J_1, \ldots, J_{2r-1}, K_0, \ldots, K_{r-2}$, input nodes $H_0, I_0, \ldots, I_{r-2}$, and output node $O$ with the following edges:

1. Nodes $H_i, J_i$ will point towards the output $O$.

2. Node $H_i$ will point towards nodes $H_{i+1}$ and $J_{i+1}$.

3. Node $K_{i-1}$ will point towards node $H_{2i}$ and node $J_{2i}$.

4. Node $I_i$ will point towards node $K_i$.

We now define the composition rules at each node:

$$\begin{aligned}
\mathfrak{R}_{H_{2i}} &= \{(h,k) \mapsto \phi(h+k)\} \\
\mathfrak{R}_{H_{2i-1}} &= \{h \mapsto Wh : \|W^\top\|_{2,1} \leq a_W, \|W\|_{\text{op}} \leq \sigma\} \text{ for } 2 \leq i \leq r-1 \\
\mathfrak{R}_{H_{2r-1}} &= \{h \mapsto Yh : \|Y^\top\|_{2,1} \leq a_Y, \|Y\|_{\text{op}} \leq \sigma_Y\} \\
\mathfrak{R}_{J_{2i}} &= \{(h,k) \mapsto D\phi[h+k]\} \\
\mathfrak{R}_{K_i} &= \{x \mapsto Ux : \|U^\top\|_{2,1} \leq a_U\}
\end{aligned}$$

Finally, nodes $J_{2i-1}$ will have composition rule $R_{J_{2i-1}} = DR_{H_{2i-1}}$. Finally, the output node $O$ will have composition rule

$$R_O(x, h_1, \ldots, h_{2r-1}, D_1, \ldots, D_{2r-1}) \triangleq$$

$$(l_\gamma(h_{2r-1}) - 1) \prod_{i=1}^{r-1} \mathbb{1}_{\leq t^{(i)}}(\|h_{2i}\|) \prod_{1\leq j<j'\leq 2r-1} \mathbb{1}_{\leq \sigma_{j'\leftarrow j}}(\|D_{j'} \cdots D_j\|_{\text{op}}) + 1$$

Note that the family of functions computed by this computation graph family is a strict superset of $\mathcal{L}_{\text{rnn-aug}}$ (as we technically allow $R_{H_{2i-1}}$, $R_{H_{2i'-1}}$ to use different matrices $W$). We will refer to this resulting family as $\tilde{\mathcal{G}}$.

First, we claim that $\tilde{G}$ satisfies the release-Lipschitz condition, with Lipschitz constants $\kappa^{\text{rnn-hidden},(i)}$ for nodes $H_{2i-1}$ and $K_{i-1}$, and $\kappa^{\text{rnn-jacobian},(i)}$ for nodes $J_{2i-1}$. (As we will see later, the Lipschitzness of nodes $V_{2i}, J_{2i}$ will not matter because the composition rules are function classes with log covering number 0.)

To see this, we note that if we release $K_0, \ldots, K_{r-2}$ from the graph and set them to fixed values, the resulting induced graph family is simply the Lipschitz augmentation of Section B for the sequential graph family on nodes $H_0, \ldots, H_{2r-1}$ and an un-augmented output. Thus, the machinery of

Theorem B.2 applies here, and we can conclude that this reduced graph family is $\kappa^{\text{rnn-hidden},(i)}$-release-Lispchitz for nodes $H_{2i-1}$ and $\kappa^{\text{rnn-jacobian},(i)}$-release-Lipschitz for nodes $J_{2i-1}$. Since this holds for any choice of $K_0, \ldots, K_{r-2}$, we can draw the same conclusion about $\tilde{\mathcal{G}}$, the augmented family that is not reduced. However, by nature of the composition rules in $\tilde{\mathcal{G}}$, the Lipschitzness of $H_{2i-1}$ and $K_{i-1}$ must be identical (as $f(x+y)$ must have the same worst-case Lipschitz constant in $x$ and $y$ for any function $f$). Thus, we get that $\tilde{G}$ satisfies release-Lipschitzness with constants $\kappa^{\text{rnn-hidden},(i)}$ for nodes $H_{2i-1}, K_{i-1}$, and $\kappa^{\text{rnn-jacobian},(i)}$ for nodes $J_{2i-1}$.

With this condition established, we can complete the proof via the same covering number argument as in Theorem C.2. $\qquad\square$

Now as in the proof of Theorem 5.1, we first observe that the augmented loss upper bounds the 0-1 classification loss, giving us a 0-1 test error bound. We then apply the same union bound technique over parameters $\gamma, t^{(i)}, \sigma_{j'\leftarrow j}, a_W, a_U, a_Y$, as in the proof of Theorem 5.1.

# H   ReLU Networks

In this section, we apply our augmentation technique to relu networks to produce a generalization bound similar to that of Nagarajan and Kolter [2019], which is polynomial in the Jacobian norms, hidden layer norms, and inverse pre-activations.

Recall the definition of neural nets in Example A.1: the neural net with parameters $\{W^{(i)}\}$ and activation $\phi$ is defined by

$$F(x) = W^{(r)}\phi(\cdots \phi(W^{(1)}x)\cdots)$$

For this section, we will set $\phi$ to be the relu activation. We also use the same notation for layers and indexing as Section 5. We first state our generalization bound for relu networks:

**Theorem H.1.** *Fix reference matrices $\{A^{(i)}\}, \{B^{(i)}\}$. With probability $1 - \delta$ over the random draws of the data $P_n$, all neural networks $F$ with relu activations parameterized by $\{W^{(i)}\}$ will have the following generalization guarantee*

$$\mathbb{E}_{(x,y)\sim P}[l_{\text{0-1}}(F(x), y)] \leq \tilde{O}\left(\frac{\left(\sum_i (\kappa^{\text{relu-hidden},(i)} a^{(i)} t^{(i-1)})^{2/3} + (\kappa^{\text{relu-jacobian},(i)} b^{(i)})^{2/3}\right)^{3/2}}{\sqrt{n}} + r\sqrt{\frac{\log(1/\delta)}{n}}\right)$$

*where*

$$
\begin{aligned}
\kappa^{\text{relu-jacobian},(i)} &\triangleq \sum_{1 \leq j \leq 2i-1 \leq j' \leq 2r-1} \frac{\sigma_{j'\leftarrow 2i}\sigma_{2i-2\leftarrow j}}{\sigma_{j'\leftarrow j}} \\
\kappa^{\text{relu-hidden},(i)} &\triangleq \frac{1}{\text{poly}(r)} + \frac{\sigma_{2r-1\leftarrow 2i}}{\gamma} + \sum_{i \leq i' < r} \frac{\sigma_{2i'\leftarrow 2i}}{t^{(i')}} + \frac{\sigma_{2i'-1\leftarrow 2i}}{\gamma^{(i')}}
\end{aligned}
\tag{35}
$$

*In these expressions, we define $\sigma_{j-1\leftarrow j} = 1$, $\gamma^{(i)}$ to be the minimum pre-activation after the $i$-th weight matrix over all coordinates in the $i$-th layer and all datapoints:*

$$\gamma^{(i)} \triangleq \min_{x\in P_n}\min_j |[F_{2i-1\leftarrow 1}(x)]_j|$$

*where $[F_{2i-1\leftarrow 1}(x)]_j$ indexes the $j$-th coordinate of $F_{2i-1\leftarrow 1}(x)$, and additionally use*

$$a^{(i)} \triangleq \text{poly}(r)^{-1} + \|W^{(i)\top} - A^{(i)\top}\|_{2,1}, b^{(i)} \triangleq \text{poly}(r)^{-1} + \|W^{(i)} - B^{(i)}\|_{1,1}$$

$$t^{(0)} \triangleq \text{poly}(r)^{-1} + \max_{x\in P_n}\|x\|, \ t^{(i)} \triangleq \text{poly}(r)^{-1} + \max_{x\in P_n}\|F_{2i\leftarrow 1}(x)\|$$

$$\sigma_{j'\leftarrow j} \triangleq \text{poly}(r)^{-1} + \max_{x\in P_n}\|Q_{j'\leftarrow j}(x)\|_{\text{op}}, \text{ and } \gamma \triangleq \min_{(x,y)\in P_n}[F(x)]_y - \max_{y'\neq y}[F(x)]_{y'} > 0$$

*Note that we assume the existence of a positive margin, so the training error here is 0.*

We note that compared to Theorem 5.1, $\kappa^{\text{relu-jacobian},(i)} = \kappa^{\text{jacobian},(i)}$, but $\kappa^{\text{relu-hidden},(i)}$ now has a dependence on the preactivations $\gamma^{(i)}$, as in Nagarajan and Kolter [2019].

We provide a proof sketch of Theorem H.1 here. We first bound the Rademacher complexity some family of augmented losses, specified precisely in Theorem H.2. The rest of the argument then follows the same way as the proof of Theorem 5.1: using Rademacher complexity to argue that the augmented losses generalize, applying the fact that the augmented losses upper-bound the 0-1 loss, and then union bounding over all choices of parameters.

**Theorem H.2.** *Following the definitions in Theorem C.2, let $\mathcal{F}$ denote the class of neural networks, $\sigma_{j'\leftarrow j}$ be parameters intended to bound the spectral norm of the $j$ to $j'$ layerwise Jacobian, and $t^{(i)}$ be parameters bounding the layer norm after applying the $i$-th activation. Define $\gamma^{(i)}$ as parameters intended to lower bound the minimum preactivations after the $i$-th linear layer. Define the class of augmented losses*

$$\mathcal{L}_{\text{relu-aug}} \triangleq \left\{ (l_\gamma - 1) \circ F \prod_{i=1}^{r-1} \mathbb{1}_{\leq t^{(i)}}(\|F_{2i\leftarrow 1}\|) \mathbb{1}_{\geq \gamma^{(i)}}(\min_j |[F_{2i-1\leftarrow 1}]_j|) \prod_{1 \leq j < j' \leq 2r-1} \mathbb{1}_{\leq \sigma_{j'\leftarrow j}}(\|Q_{j'\leftarrow j}\|_{\text{op}}) + 1 : F \in \mathcal{F} \right\}$$

*where $\mathbb{1}_{\geq \gamma^{(i)}} \triangleq 1 - \mathbb{1}_{\leq \gamma^{(i)}}/2$. Define for $1 \leq i \leq r$, $\kappa^{\text{relu-jacobian},(i)}$, $\kappa^{\text{relu-hidden},(i)}$ meant to bound the influence of the matrix $W^{(i)}$ on the Jacobians and hidden variables, respectively, as in (35). Then the augmented loss class $\mathcal{L}_{\text{relu-aug}}$ has empirical Rademacher complexity upper bound*

$$\text{Rad}_n(\mathcal{L}_{\text{relu-aug}}) = \tilde{O}\left( \frac{\left(\sum_i (\kappa^{\text{relu-hidden},(i)} a^{(i)} t^{(i-1)})^{2/3} + (\kappa^{\text{relu-jacobian},(i)} b^{(i)})^{2/3}\right)^{3/2}}{\sqrt{n}} \right)$$

Note the differences with Theorem C.2: the augmented loss class $\mathcal{L}_{\text{relu-aug}}$ now includes the additional indicators $\mathbb{1}_{\geq \gamma^{(i)}}(\min_j |[F_{2i-1\leftarrow 1}]_j|)$, and we use the Lipschitz constants $\kappa^{\text{relu-hidden},(i)}$, $\kappa^{\text{relu-jacobian},(i)}$ defined in Theorem H.1.

*Proof sketch.* As in the proof of Theorem C.2, associate the loss class $\mathcal{L}_{\text{relu-aug}}$ with a family $\widetilde{\mathcal{G}}$ of computation graphs on internal nodes $V_1, \ldots, V_{2r-1}, J_1, \ldots, J_{2r-1}$ as follows: define the graph structure to be identical to the Lipschitz augmentation of a sequential computation graph family (Figure 3) and define the composition rules

$$\mathfrak{R}_{V_{2i}} = \{\phi\}$$

$$\mathfrak{R}_{V_{2i-1}} = \{h \mapsto Wh : \|W^\top - A^{(i)^\top}\|_{2,1} \leq a^{(i)}, \|W - B^{(i)}\|_{1,1} \leq b^{(i)}, \|W\|_{\text{op}} \leq \sigma^{(i)}\}$$

Assign to the $J_i$ nodes composition rule $R_{J_i} = DR_{V_i}$, and finally, assign to the output node $O$ the composition rule

$$R_O(x, v_1, \ldots, v_{2r-1}, D_1, \ldots, D_{2r-1}) \triangleq$$

$$(l_\gamma(v_{2r-1}) - 1) \prod_{i=1}^{r-1} \mathbb{1}_{\leq t^{(i)}}(\|v_{2i}\|) \mathbb{1}_{\geq \gamma^{(i)}}(\min_j |[v_{2i-1}]_j|) \prod_{1 \leq j \leq j' \leq 2r-1} \mathbb{1}_{\leq \sigma_{j'\leftarrow j}}(\|D_{j'}\cdots D_j\|_{\text{op}}) + 1$$

The resulting family of computation graphs will compute $\mathcal{L}_{\text{relu-aug}}$. Now we claim that $\widetilde{\mathcal{G}}$ is $\kappa^{\text{relu-hidden},(i)}$-release-Lipschitz in nodes $V_{2i-1}$ and $\kappa^{\text{relu-jacobian},(i)}$-release-Lipschitz in nodes $J_{2i-1}$. (Note that the Lipschitzness of nodes $V_{2i}, J_{2i}$ will not matter because the associated function classes and singletons and therefore have a log covering number of 0 anyways).

The argument for the $\kappa^{\text{relu-jacobian},(i)}$-release-Lipschitzness of $J_{2i-1}$ follows analogously to the argument of Lemma F.8 and Theorem C.2.

To see the $\kappa^{\text{relu-hidden},(i)}$-release-Lipschitzness of $V_{2i-1}$, we first note that we can account for the instantaneous change in the graph output given a change to $V_{2i-1}$ as a sum of the following: 1) the change in $l_\gamma(V_{2r-1}) - 1$ multiplied by the other indicators, 2) the change in the term $\mathbb{1}_{\leq t^{(i')}}(\|V_{2i}\|)\mathbb{1}_{\geq \gamma^{(i)}}(\min_j |[V_{2i-1}]_j|)$ multiplied by the other indicators, and 3) the change in $\mathbb{1}_{\leq \sigma_{j'\leftarrow j}}(\|J_{j'}\cdots J_j\|_{\text{op}})$ multiplied by the other indicators. The term 1) can be computed as $\frac{\sigma_{2r-1\leftarrow 2i}}{\gamma}$, term 2) can be accounted for by $\frac{\sigma_{2i'\leftarrow 2i}}{t^{(i')}} + \frac{\sigma_{2i'-1\leftarrow 2i}}{\gamma^{(i')}}$, and finally the term 3) is 0 because as relu is piecewise-linear, the instantaneous change in the Jacobian is 0 if all preactivations are bounded

away from 0, and in the case that the preactivations are not bounded away from 0, the indicator $\mathbb{1}_{\geq \gamma^{(i)}}(\min_j |[V_{2i-1}]_j|)$ takes value 0. The same steps as Lemma F.1 can be used to formalize this argument.

Finally, to conclude the desired Rademacher complexity bounds given the release-Lipschitzness, we apply the same reasoning as in Theorem C.2. □

# I  Additional Experimental Details

## I.1  Implementation Details for Jacobian Regularizer

For all settings, we train for 200 epochs with learning rate decay by a factor of 0.2 at epochs 60, 120, and 150. We additionally tuned the value of $\lambda$ from values $\{0.1, 0.05, 0.01\}$ for each setting: for the experiments displayed in Figure 1, we used the following values:

1. Low learning rate: $\lambda = 0.1$
2. No data augmentation: $\lambda = 0.1$
3. No BatchNorm: $\lambda = 0.05$

For all other hyperparameters, we use the defaults in the PyTorch WideResNet implementation: `https://github.com/xternalz/WideResNet-pytorch`, and we base our code off of this implementation. We report results from a single run as the improvement with Jacobian regularization is statistically significant. We train on a single NVIDIA TitanXp GPU.

## I.2  Empirical Scaling of our Complexity Measure with Depth

In this section, we empirically demonstrate that the leading term of our bounds can exhibit better scaling in depth than prior work.

Figure 5: Log leading terms for spectral vs. our bound on WideResNet trained on CIFAR10 using different depths.

We compute leading terms of our bound: $\frac{\sum_i \max_{x \in P_n} \|h^{(i)}(x)\|_2 \max_{x \in P_n} \|J^{(i)}(x)\|\|_{\text{op}}}{\gamma}$, where $i$ ranges over the layers, $h^{(i)}$, $J^{(i)}$ denote the $i$-th hidden layer and Jacobian of the output with respect to the $i$-th hidden layer, respectively, and $\gamma$ denotes the smallest positive margin on the training dataset. We compare this quantity with that of the bound of [Bartlett et al., 2017]: $\prod_i \|W^{(i)}\|_{\text{op}}/\gamma$. In Figure 5, we plot this comparison for WideResNet[6] models of depths 10, 16, 22, 28 trained on CIFAR10. For all models, we remove data augmentation to ensure that our models fit the training data perfectly. We train each model for 50 epochs, which is sufficient for perfectly fitting the training data, and start from an initial learning rate of 0.1 which we decrease by a factor of 10 at epoch 30. All other parameters are set to the same as their defaults in the PyTorch WideResNet implementation: `https://github.com/xternalz/WideResNet-pytorch`. We plot the final complexity measures computed on a single model. We note that our models are trained with Batchnorm. At test time,

these Batchnorm layers compute affine transformations, so we compute the bound by merging these transformations with the adjacent linear layer.

Figure 5 demonstrates that our complexity measure can be much lower than the spectral complexity. Furthermore, in Figure 5, our complexity measure appears to scale well with depth for WideResNet models.

## J    Toolbox

**Claim J.1.** *Consider the function* $u : [0, 1] \times [0, 1] \mapsto \mathbb{R}$ *defined as follows:* $u(x_1, x_2) = (x_1 - 1)x_2 + 1$. *Then the following statements hold:*

1. *The function* $u$ *outputs values in* $[0, 1]$.

2. $u(x_1, x_2) \geq x_1$.

3. $u(u(x_1, x_2), x_3) = u(x_1, x_2 x_3)$.

*Proof.* First, we note that $u(x_1, x_2) = x_1 x_2 + 1 - x_2 \leq x_2 + 1 - x_2 = 1$. Furthermore, $u(x_1, x_2) \geq x_1 x_2 + x_1(1 - x_2) = x_1$, which completes the proof of statements 1 and 2. To prove the third statement, we note that $u(u(x_1, x_2), x_3) = (x_1 x_2 + 1 - x_2)x_3 + 1 - x_3 = x_1 x_2 x_3 + 1 - x_2 x_3 = u(x_1, x_2 x_3)$. $\square$

**Claim J.2** (Chain rule Wikipedia contributors [2019]). *The Jacobian of a composition of a sequence of functions* $f_1, \ldots, f_k$ *satisfies*

$$Df_{k \leftarrow 1}(x) = Df_k(f_{(k-1) \leftarrow 1}(x)) \cdot Df_{k-1}(f_{(k-2) \leftarrow 1}(x)) \cdots Df_2(f_1(x)) \cdot Df_1(x) \quad (36)$$

*where the $\cdot$ notations are standard matrix multiplication. For simplicity, we also write in the function form:*

$$Df_{k \leftarrow 1} = (Df_k \circ f_{(k-1) \leftarrow 1}) \cdot (Df_{k-1} \circ f_{(k-2) \leftarrow 1}) \cdots (Df_2 \circ f_1) \cdot Df_1 \quad (37)$$

**Claim J.3** (Telescoping sum). *Let* $p_1, \ldots, p_m$ *and* $q_1 \ldots q_m$ *be two sequence of functions from* $\mathbb{R}^d$ *to* $\mathbb{R}$. *Then,*

$$p_1 p_2 \cdot p_m - q_1 q_2 \cdot q_m = (p_1 - q_1)p_2 \cdots p_m + q_1(p_2 - q_2)p_3 \cdots p_m + \cdots + q_1 \cdots q_{m-1}(p_m - q_m) \quad (38)$$

**Claim J.4** (Bounding function differences). *Let* $f : \mathcal{D} \to \mathcal{D}'$, *and consider the total derivative* $Df$ *operator mapping* $\mathcal{D}$ *to a linear operator between normed spaces* $\mathcal{D}$ *to* $\mathcal{D}'$. *Suppose that* $Df[x]$ *is* $\kappa$-*Lipschitz in* $x$, *in the sense that* $\|Df[x] - Df[x + \nu]\|_{\mathrm{op}} \leq \kappa \|\nu\|$, *where* $\| \cdot \|_{\mathrm{op}}$ *is the operator norm induced by* $\mathcal{D}$ *and* $\mathcal{D}'$. *Then*

$$\|f(x) - f(x + \nu)\| \leq (\|Df[x]\|_{\mathrm{op}} + \frac{\kappa}{2}\|\nu\|)\|\nu\| \quad (39)$$

*Furthermore,*

$$\|f(x) - f(x + \nu)\| \mathbb{1}_{\leq \tau_f}(\|Df[x]\|_{\mathrm{op}}) \leq (2\tau_f + \frac{\bar{\tau}}{2}\|\nu\|)\|\nu\| \quad (40)$$

*Proof.* We write $f(x + \nu) - f(x) = \left(\int_{t=0}^1 Df[x + t\nu]dt\right)\nu$. Now we note that

$$\|\int_{t=0}^1 Df[x + t\nu]dt\|_{\mathrm{op}} \leq \int_{t=0}^1 \|Df[x + t\nu]\|_{\mathrm{op}}dt \qquad \text{(by triangle inequality)}$$

$$\leq \int_{t=0}^1 (\|Df[x]\|_{\mathrm{op}} + t\kappa\|\nu\|)dt \qquad \text{(by Lipschitzness of } Df)$$

$$\leq \|Df[x]\|_{\mathrm{op}} + \frac{\kappa}{2}\|\nu\| \quad (41)$$

Thus,

$$\|f(x + \nu) - f(x)\| \leq \|\int_{t=0}^1 Df[x + t\nu]dt\|_{\mathrm{op}}\|\nu\|$$

$$\leq \left(\|Df[x]\|_{\mathrm{op}} + \frac{\kappa}{2}\|\nu\|\right)\|\nu\| \qquad \text{(by (41))}$$

which proves (39).

To prove (40), we consider two cases.first, if $\|Df[x]\|_{\mathrm{op}} > 2\tau_f$, then $\mathbb{1}_{\leq \tau_f}(\|Df[x]\|_{\mathrm{op}}) = 0$ so (40) immediately holds. Otherwise, if $\|Df[x]\|_{\mathrm{op}} \leq 2\tau_f$, we can plug this into (39) to obtain (40), as desired. □