[Reviews · NeurIPS 2019]

Reviewer 1



Summary. Generalization bounds on neural nets, based on Rademacher complexity, use the norm bounds on weights of layers, which gives an exponential dependence on depth. Moreover, existing lower bounds show that this is unavoidable (in general). The goal of the paper is to get bounds polynomial in depth by additionally using properties of training data. However, such data dependent bounds comes with challenges, discussed in the paper. The authors introduce "augmenting" the loss function with desirable properties and present tools to derive covering bounds on augmented loss. Comments: 1. Data-dependent generalization bounds have recently become popular to derive sharper generalization bounds. This paper contributes to this line of work by considering properties of training data, in particular norms of layers and norms of Jacobians of laters with other layers. They paper presents the (novel) idea of augmenting the loss function with the desirable properties, they then derive generalization bounds on the augmented loss. 2. The paper is technically rigorous and the authors develop a lot of tools, in broad generality, which can be helpful in other related problems. 3. Even though the paper is technically heavy with a lot of notation, I found that it was rather well written, with all the main ideas well presented and explained. The authors present the idea in more generality than perhaps needed (which complicates notation perhaps) but I think it would be difficult to simplify. 3. The authors present the abstraction of computational graph (which computes the loss of network) and show that "augmenting" the graph in a way corresponds to augmenting the loss. They then develop covering bounds on the augmented loss. The main technical contribution is augmenting the loss functions to enforce desirable properties, and also deriving covering bounds. The technical challenges encountered are well described, although it becomes much clear after reading appendix A and B rather than the main paper (more on this below), 4. The presentation gets perhaps very dense in the technical sections, and unable to put forth the key ideas. In particular, the machinery of computational graph, although connected to neural networks, looked overkill to me at first. I then went through the corresponding sections A and B in the appendix to understand it well. While it is understandable that the authors kept a condensed version due to page restrictions, fleshing out sections 5 and 6 better can help accessibility to a wider audience. 5. The results in the paper apply to smooth activations and therefore not apply to ReLU networks which are (arguably) the widely used ones in practice. It would be nice if the authors could discuss the challenges to extending it to ReLU networks. 6. While the main idea of the paper is reiterated: removing exponential dependence on depth, it would be helpful to see how the bound compares to existing results on simple datasets(?). The only experiment they provide in the appendix is on regularizing with respect to their complexity improves performance. It would be more insightful to compare against existing bounds and show (as claimed) that with increasing depth, the upper bound is more meaningful than previous work. ------------ I thank the authors for the clarifications ; I have read them and would keep my score as it is.

Reviewer 2



This paper studies data-dependent generalization bounds in multi-layer feedforward neural networks with smooth activation functions. The authors give a new Rademacher complexity bound, which is polynomial in the hidden layer norms and the interlayer Jacobian norms. Assuming these empirical quantities are polynomial in the depth, the Rademacher complexity bound obtained is polynomial in the depth, while a naïve Rademacher complexity bound (product of layer matrix norms) is exponential in the depth. In the experiment, the authors explicitly regularize the networks’ Jacobians and improve test error. In the proof, the authors bound the empirical Rademacher complexity by bounding the covering number of the function class. To exploit the data-dependent properties, the authors augment the original loss by including indicators functions which encode the properties. In order to bound the covering number of the augmented function, the authors develop a general theory to bound the covering number of a computational graph. Basically, the authors show that as long as the release-Lipschitzness condition holds and the internal nodes are bounded, we can cover a family of computational graphs by covering the function class at each vertex. The authors further design an augmentation scheme for any sequential computational graph such that the augmented graph is release-Lipschitz. Overall this is a good theory paper which studies the data-dependent generalization in neural networks. My only concern is on the assumption that the interlayer Jacobian is small. It might be better to identify some scenarios in which we can prove that these terms are indeed small (polynomial in depth). At least, the authors can include more intuitions to explain why the interlayer Jacobian should be small in practice. Some minor comment: Line 246: “Let R_V the collection” -> “Let R_V be the collection” ------------------------------------------------------- I have read the authors' rebuttal and other reviews. I will keep my score.

Reviewer 3



I've read the author response and updated my score accordingly. I thank the authors for providing answers for clarification. As for the width-dependence, I still think it is useful to have some plots in the paper for varying widths, with fixed depth, regardless of any lack of pattern in these plots. Similarly, it'll be useful to have a discussion on the worst-case and best-case dependence of these norms on the width. Regarding my question in point 5, thanks for clarifying! I completely missed the "for all x" in the statement, my mistake (All I saw was an upper bound that was quadratic in \nu, and I was surprised how it resulted in a bound on the Lipschitz constant everywhere.) Besides this, I hope that authors incorporate the suggestions, and if possible, also spend more time on the appendix to make the discussion easier, in whatever way possible. ====== Summary: This paper presents a Rademacher complexity/covering number based generalization bound for deep networks that does not depend on the worst-case approximation of the Lipschitz constant of the network: the product of the spectral norms of the weight matrices (which is exponential in depth). This dependence is instead replaced by a dependence on the Jacobians of the network which are arguably nicer terms that represent the interactions between the weight matrices more intricately. While most works have achieved such bounds only on a modified network, this paper manages to derive on the original network learned by SGD. In order to do this, the paper develops an extensive set of graph-like abstractions (to modularize the proofs), auxiliary tools and also most importantly, an "augmented loss" as the main subject of all their analysis. (The augmented loss is the classification loss combined with losses that correspond to whether the data-dependent norms are bounded or not.) Using these tools, the paper shows how the Lipschitz constant of the augmented loss can be bounded nicely, and how this results in a tighter covering number bound on a space of networks. 1. Significance: Developing spectral-norm-independent bounds (or more generally, bounds tighter in some asymptotic sense) on the *original* network learned by SGD (without subjecting the network to any modification that is heuristic or otherwise) is an important and challenging problem that I believe needs more attention in the community. Hence, I appreciate the goal of this paper. 2. Novelty: Although the above goal in itself is not novel, there are no adequate tools to achieve this goal for Rademacher complexity based bounds. The paper carefully develops a complex and novel framework to achieve this -- these ideas and the technical challenges within are very different from the PAC-Bayes counterpart that was proposed in Nagarajan and Kolter '19. Although I did not deeply engage with the proofs in the (long) appendix, I did gain a high level idea of what is being done in this paper. The main challenge in both this paper and in N&K'19 is that, while deriving the generalization bound, one must be rigorous in incorporating the Jacobian norms, which depend on the input to the network. Since the generalization bound itself cannot depend on any norms that depend on unseen data, the whole analysis should somehow rely only on the Jacobian norms computed on training data. While N&K '19 achieve this by generalizing these norms from training to test data, this paper achieves this by defining an augmented loss whose Lipschitz constant can be tightly bounded for _any_ input. If I understand this right, this is a key step in their proof (Lemma G.1 specifically), and this is clever. Furthermore, this method subsumes that of N&K '19, as has been established in Theorem 1.1. 3. The tools developed here are pretty general, hence I believe this leaves open further extension/research in this space. 4. Clarity: As stated above, the tools required to do this are very complex and fundamentally new. Hence they require their own grammar and abstractions to be presented rigorously & succinctly. Although I did not do a deep reading of all of this, I do think that a lot of care has been taken to develop these terminologies & notations. The paper is also very well-written, with the essence of the ideas presented neatly in the main paper. Having said this, I do think that absorbing the ideas in this paper will generally take more time than most theory papers, but the blame here would mostly lie on the fact that the ideas are new. Some important questions: ======== 5. I'm somewhat bothered by the fact that the Lipschitz constant of the augmented loss is bounded only for small perturbations of the parameters (see lines 851, 859). (I see that this is needed because the change in the loss grows quadratically with the perturbation in the parameters). Shouldn't the upper bound on \nu in line 859 turn up somewhere else in the proof? If I understand it right, it should somehow affect the application of the Dudley's entropy theorem in line 687 (specifically, in how the integral simplifies). Perhaps I'm missing some simple algebraic step here, but I would really like a clarification here. 6. I'd like to verify my understanding of how the high level approach here relates to that in the PAC-Bayesian framework of N&K '19. The idea in N&K was to bound, layer-by-layer, the proportion of test data on which the Jacobian norms are too large, by making use of the Jacobian norms on the training data. More generally, it seems natural that any approach which achieves this sort of a data-dependent bound would necessarily have to generalize these norm bounds from the training data to test data in some way or the other. This gives rise to the question, which I think may be worth discussing in the paper (maybe in the appendix): precisely which part of the proof here makes a similar argument about the Jacobian norms on arbitrary unseen inputs? Like I expressed in point 2, I suspect this is implicitly taken care of by Lemma G.1 which argues that the Lipschitz constant of the augmented loss is bounded nicely for any input. But I'd appreciate a short clarification about this from the authors. ======= Some important suggestions: ====== 7. For the sake of transparency, I strongly suggest a more prominent disclaimer with regards to the claim about the tighter bounds: that it holds a) only for smooth activations and hence, b) not for ReLU activations (in which case the bound suffers from dependence on the inverse pre-activations). Most papers in this space focus on ReLU networks, and hence this is an important distinction to make while making a positive claim. Of course, the paper does specify this somewhere in the introduction, but this detail is an crucial one that can be easily overlooked by a casual reader. I suggest that this be highlighted in the (a) abstract itself and (b) emphasized in bold/italics in line 42 and again (c) in lines 66-70 for a fair comparison with prior work. 8. I also think it's important to clearly state the parameter-count dependencies in these bounds, especially because these bounds are very complicated. This will enable future work to better compare their results with those in the paper. Specifically, I think it's useful to (a) have a theoretical discussion on the worst/best/expected dependence of the terms in this bound on width & depth and (b) empirical observations of the same. (I am not concerned with how bad/good the exact dependence on depth/width since I'm already happy with the fact that there's an exponential improvement in terms of depth). 9. I believe Theorem D.2 can be more rigorously stated. - it's not specified whether the Jacobian-bounds, \sigma are computed only on the training set P_n, although this is stated to be the case for the layer norm bounds, t. - In addition to this, I'm also a bit confused by the presence of the training set P_n in Theorem D.2. My understanding (maybe, misunderstanding) is that in Theorem D.2, there's no notion of a training set as such. \sigma, \kappa etc., are some constants which are used to define the augmented loss in the context of that theorem, and the theorem bounds the Rademacher complexity of this augmented loss. And later on, as stated in line 697/698, a union bound is applied on all possible choices of these parameters, and the final generalization bound is instantiated for sigma and kappa set to something close to what was observed on the training set P_n -- and this is where the training set should appear, and not in Theorem D.2. Minor suggestions: ===== - It took me a few minutes to parse the augmented loss. It wasn't immediately clear why there is a "-1" and a "+1" that have been separated out. I think a one-line verbal description of this would be useful. - It'd be useful to define covering numbers formally in the appendix, for completeness. - In the beginning of Section of 5 and 6, it'd be helpful to remind the reader that these are essentially generalizations of the discussions in Section 4. Minor Corrections: ==== In Proposition D.1, an epsilon must be changed to alpha. (Also, why is this a proposition?) Line 185 can easily scales Line 196 - soft indicators Lien 84: it was not immediately clear what this form meant Line 211: augmentation Line 244: generalizes Line 726 is true IF O is only... Line 640: isn't r missing before the log term? Line 828: function real valued function ===

[Author Response · NeurIPS 2019]

We thank the reviewers for the detailed and insightful reviews. As noted by the reviewers, our work 1) develops
generalization bounds for neural nets with improved depth dependency, and 2) "contributes novel tools and techniques
for generalization theory." We answer most of the questions and will incorporate the feedbacks into the final version.

**[R1,R4] Empirical Comparison of Bounds:**

Figure 1: Left: Log generalization bounds (without $1/\sqrt{n}$) for fully connected networks trained on MNIST vs. depth.
Right: Log leading terms for spectral vs. our bound on WideResNet trained on CIFAR10 using different depths.

In Figure 1, we address questions about empirical evaluation of our bounds. First, on the left we train a fully-connected
neural net on MNIST and measure the our bound and bounds in previous papers.[1] The plot indicates that our bound is
generally several magnitudes smaller than existing bounds based on product of matrix norms. Second, on the right we
compare leading terms of our bound for WideResNet: $\frac{\sum_i \max_{x \in P_n} \|h_i(x)\|_2 \max_{x \in P_n} \|J_i(x)\|_{op}}{\gamma}$[2], where $i$ ranges over the
layers, with that of the bound of [Bartlett et al., 2017]: $\prod_i \|W^{(i)}\|_2/\gamma$. We note that our complexity measure scales
better with depth for WideResNet than fully-connected nets, which indicates that architecture influences Jacobian and
hidden layer norms.

**[R1]:** "It would be nice if the authors could discuss the challenges to extending it to ReLU networks."
• The primary challenge is that Theorem 5.1 requires the augmented indicators on the Jacobian norms to be themselves
Lipschitz w.r.t. the hidden layers. However, the Jacobians of relu networks are piecewise constant and therefore not
Lipschitz – thus, to control the change in the Jacobians, we must condition that the pre-activations are bounded away
from 0, leading to the dependency on inverse pre-activations in Nagarajan and Kolter [2019] and our Theorem I.1.

**[R2]:** "It might be better to identify some scenarios in which we can prove that [Jacobians] are indeed small (polynomial
in depth) ... the authors can include more intuitions to explain why the interlayer Jacobian should be small in practice."
• It seems to be a very challenging open question to rigorously prove that the Jacobian norms will be small (as this
would require characterizing the solution obtained by gradient descent), but many empirical findings support that the
interlayer Jacobian on training data should be small in practice compared to the product of spectral norms (e..g, [Arora
et al., 2018, Figure 1], Nagarajan and Kolter [2019], Novak et al. [2018]). Intuitively speaking, for linearized deep
nets, the Jacobian norm is of the form $\|W^{(k)} \cdots W^{(j)}\|_{op}$, which could be much smaller than $\|W^{(k)}\|_{op} \cdots \|W^{(j)}\|_{op}$
when there is cancellation within the weights.

**[R4]:** "Shouldn't the upper bound on $\nu$ in line 859 turn up somewhere else in the proof?"
• We implicitly used the following fact: suppose $\exists \delta > 0$, such that $\forall x, y$ with $\|x - y\| \leq \delta$, $|\tilde{z}(x) - \tilde{z}(y)| \leq \tau \|x - y\|$,
then, $|\tilde{z}(x) - \tilde{z}(y)| \leq \tau \|x - y\|$ is true for any $x, y$. This can be proven by dividing the line between $x$ and $y$ into
segments of length $\delta$ and applying the given property on each segment. (We will clarify more in the next revision.)

**[R4]:** "precisely which part of the proof here makes a similar argument about the Jacobian norms on arbitrary unseen
inputs?... I suspect this is implicitly taken care of by Lemma G.1"
• The intuition is correct – the same argument as Lemma G.1 shows that the product of indicators on Jacobian and
hidden layer norms is globally Lipschitz, and therefore the the product of indicators generalizes to unseen inputs.

**[R4]:** "(a) have a theoretical discussion on the worst/best/expected dependence of the terms in this bound on width &
depth and (b) empirical observations of the same."
• Our bounds have explicit polynomial dependence on depth, but no explicit dependence on width – they instead
depend on the $(2, 1)$ and $(1, 1)$ weight matrix norms. Theoretically, this is comparable to previous bounds [Bartlett
et al., 2017, Neyshabur et al., 2017], which also depend on such norms. Empirically, for deep networks it's unclear how
these bounds depend on width in practice. (Neyshabur et al. [2018] study this but for shallow depth 2 networks.)

**[R4]:** "confused by ... training set $P_n$ in Thm D.2. My understanding ... there's no notion of a training set as such..."
• Your understanding is exactly correct - $\sigma, t$ are arbitrary parameters in Theorem D.2 and only set to their values
computed on $P_n$ when we apply the union bound of line 697/698. We will clarify this point in future revisions.

## Footnotes

[1]Experimental details: We use BatchNorm layers (to allow training with larger depth) with a hidden layer width of 40 on 1000 randomly sampled images from MNIST. At test time, BatchNorm is an affine transformation,so we merge it into the adjacent linear layer and then compute the bound. We restrict the training set for faster computation of bounds and easier optimization.

[2]We compute leading terms only for faster computation. $h_i$ denotes hidden layer $i$. $J_i$ is the output Jacobian w.r.t. to layer $i$. Our bound as stated in the paper technically does not apply to ResNet because the skip connections complicate the Lipschitz augmentation step. This can be remedied with a slight modification to our augmentation step, which we omitted for simplicity.


[Meta-Review · NeurIPS 2019]

This paper developed a new generalization error bounds for smooth activation deep neural networks which is norm-based and has only polynomial dependence on the depth unlike (most of) previous work. The bound is tighter than the ever obtained ones in the same direction, and gives new insight on the generalization error analysis for the deep learning. In particular, it connects the deep and shallow network analysis more appropriately than existing researches. The paper would benefit from more intuitive expositions of the bound and more comparisons with existing bounds on real datasets.